# EFFICIENT CONTINUAL FINITE-SUM MINIMIZATION

**Ioannis Mavrothalassitis,**\* **Stratis Skoulakis,**\* **Leello Tadesse Dadi, Volkan Cevher**
LIONS, École Polytechnique Fédérale de Lausanne
{ioannis.mavrothalassitis, efstratios.skoulakis, leello.dadi, volkan.cevher}@epfl.ch

## ABSTRACT

Given a sequence of functions $f_1, \ldots, f_n$ with $f_i : \mathcal{D} \mapsto \mathbb{R}$, finite-sum minimization seeks a point $x^\star \in \mathcal{D}$ minimizing $\sum_{j=1}^n f_j(x)/n$. In this work, we propose a key twist into the finite-sum minimization, dubbed as *continual finite-sum minimization*, that asks for a sequence of points $x_1^\star, \ldots, x_n^\star \in \mathcal{D}$ such that each $x_i^\star \in \mathcal{D}$ minimizes the prefix-sum $\sum_{j=1}^i f_j(x)/i$. Assuming that each prefix-sum is strongly convex, we develop a first-order continual stochastic variance reduction gradient method (CSVRG) producing an $\epsilon$-optimal sequence with $\tilde{\mathcal{O}}(n/\epsilon^{1/3} + 1/\sqrt{\epsilon})$ overall *first-order oracles* (FO). An FO corresponds to the computation of a single gradient $\nabla f_j(x)$ at a given $x \in \mathcal{D}$ for some $j \in [n]$. Our approach significantly improves upon the $\mathcal{O}(n/\epsilon)$ FOs that StochasticGradientDescent requires and the $\mathcal{O}(n^2 \log(1/\epsilon))$ FOs that state-of-the-art variance reduction methods such as Katyusha require. We also prove that there is no natural first-order method with $\mathcal{O}\left(n/\epsilon^\alpha\right)$ gradient complexity for $\alpha < 1/4$, establishing that the first-order complexity of our method is nearly tight.

## 1 INTRODUCTION

Given $n$ data points describing a desired input and output relationship of a model, a cornerstone task in supervised machine learning (ML) is selecting the model's parameters enforcing data fidelity. In optimization terms, this task corresponds to minimizing an objective with the *finite-sum structure*.

**Finite-sum minimization:** Given a sequence of functions $f_1, \ldots, f_n$ with $f_i : \mathcal{D} \mapsto \mathbb{R}$, let $x^\star \in \arg\min_x g(x) := \sum_{i=1}^n f_i(x)/n$. Given an accuracy target $\epsilon > 0$, *finite-sum minimization* seeks an approximate solution $\hat{x}$, such that:

$$g(\hat{x}) - g(x^\star) \leq \epsilon \tag{1}$$

we call such an $\hat{x}$ an $\epsilon$-accurate solution. If $\hat{x}$ is random then (1) takes the form $\mathbb{E}\left[g(\hat{x})\right] - g(x^\star) \leq \epsilon$.

In contemporary machine learning applications, $n$ can be in the order of billions, which makes it clear that methods tackling finite-sum minimization must efficiently scale with $n$ and the accuracy $\epsilon > 0$.

First-order methods have been the standard choice for tackling (1) due to their efficiency and practical behavior. The complexity of a first-order method is captured through the notion of overall *first-order oracles* (FOs) where a first-order oracle corresponds to the computation of a single gradient $\nabla f_i(x)$ for some $i \in [n]$ and a point $x \in \mathcal{D}$. In case each function $f_i$ is strongly convex, Stochastic Gradient Descent requires $\mathcal{O}(1/\epsilon)$ FOs Robbins and Monro (1951); Kiefer and Wolfowitz (1952) so as to solve (1). Over the last years, the so-called variance reduction methods are able to solve strongly convex finite-sum problems with $\mathcal{O}(n \log(1/\epsilon))$ FOs Nguyen et al. (2017b); Johnson and Zhang (2013); Xiao and Zhang (2014); Defazio et al. (2014); Roux et al. (2012b); Allen-Zhu (2017).

Continual Finite-Sum Minimization: In many modern applications new data constantly arrives over time. Hence it is important that a model is constantly updated so as to perform equally well both on the past and the new data Castro et al. (2018); Rosenfeld and Tsotsos (2018); Hersche et al. (2022). Unfortunately updating the model only with respect to the new data can lead to a vast deterioration of its performance over the past data. The latter phenomenon is known in the context of *continual learning* as *catastrophic forgetting* Castro et al. (2018); Goodfellow et al. (2014); Kirkpatrick et al. (2017); McCloskey and Cohen (1989). At the same time, retraining the model from scratch using

---

\*Equal contribution

both past and new data comes with a huge computational burden. Motivated by the above, we study a twist of (1), called *continual finite-sum minimization*.

**Continual Finite-Sum Minimization:** Given a sequence of functions $f_1, \ldots, f_n$ with $f_i : \mathcal{D} \mapsto \mathbb{R}$, let $x_i^\star \in \arg\min_{x \in \mathcal{D}} g_i(x) := \sum_{j=1}^{i} f_j(x)/i$ for all $i \in [n]$. Given an accuracy target $\epsilon > 0$, *continual finite-sum minimization* seeks a sequence of approximate solutions $\hat{x}_1, \ldots, \hat{x}_n \in \mathcal{D}$, such that:

$$g_i(\hat{x}_i) \leq g_i(x_i^\star) + \epsilon \quad \text{for each } i \in [n] \tag{2}$$

We call such a sequence an $\epsilon$-optimal sequence. If $x_i^\star$ is random then (2) takes the form $\mathbb{E}[g_i(\hat{x}_i)] - g_i(x_i^\star) \leq \epsilon$

**Remark 1.** *Throughout the paper we assume that each function $f_i(\cdot)$ is strongly convex, smooth and that $\mathcal{D}$ is convex and bounded. See Section 2 for the exact definitions.*

Notice that $f_i(\cdot)$ can capture a new data point meaning that (2) guarantees that, the model (parameterized by $\hat{x}_i$) is well-fitted in all of the data seen so far for each stage $i \in [n]$. As already discussed, since $n$ can be in the order of millions, it is important to design first-order methods for continual finite-sum minimization that efficiently scale with $n$ and $\epsilon > 0$ with respect to overall FOs.

A first attempt for tackling (2) is via existing first-order methods for finite-sum minimization. As discussed above StochasticGradientDescent can guarantee accuracy $\epsilon > 0$ for (1) using $\mathcal{O}(1/\epsilon)$. As a result, at each stage $i \in [n]$ one could perform $\mathcal{O}(1/\epsilon)$ stochastic gradient decent steps to guarantee accuracy $\epsilon > 0$. However the latter approach would require overall $\mathcal{O}(n/\epsilon)$ FOs. On the other hand one could use a VR methods to select $\hat{x}_i \in \mathcal{D}$ at each stage $i \in [n]$. As already mentioned such methods require $\mathcal{O}(i \log(1/\epsilon))$ FOs to guarantee accuracy $\epsilon > 0$ at stage $i \in [n]$. Thus the naive use of a VR method such as SVRG Johnson and Zhang (2013); Xiao and Zhang (2014), SAGA Defazio et al. (2014), SARAH Nguyen et al. (2017b) and Katyusha Allen-Zhu (2017) would require overall $\mathcal{O}(n^2 \log(1/\epsilon))$ FOs.

We remark that for large values of $n$ and even mediocre accuracy $\epsilon > 0$ both $\mathcal{O}(n/\epsilon)$ and especially $\mathcal{O}(n^2 \log(1/\epsilon))$ are prohibitely large. As a result, the following question arises:

**Question 1.** *Are there first-order methods for continual finite-sum minimization whose required number of FOs scales more efficiently with respect to $n$ and $\epsilon > 0$?*

**Our Contribution** The main contribution of this work is the design of a first-order method for Continual finite-sum minimization, called CSVRG, with overall $\tilde{\mathcal{O}}\left(n/\epsilon^{1/3} + \log n/\sqrt{\epsilon}\right)$ FO complexity. The latter significantly improves upon the $\mathcal{O}(n/\epsilon)$ FO complexity of SGD and the $\mathcal{O}(n^2 \log(1/\epsilon))$ complexity of variance reduction methods. As a naive baseline, we also present a *sparse* version of SGD tailored (2) with $\mathcal{O}(\log n/\epsilon^2)$ FO complexity. The latter improves on the $n$ dependence of SGD with the cost of a worse dependence on $1/\epsilon$.

Since variance-reduction methods such as Katyusha are able to achieve $\mathcal{O}(n^2 \log(1/\epsilon))$ FO complexity. A natural question is whether there exists a first-order method maintaining the $\log(1/\epsilon)$ dependence with a *subquadratic dependence* on $n$, $\mathcal{O}\left(n^{2-\alpha} \log(1/\epsilon)\right)$. We prove that the latter goal cannot be achieved for a wide class of first-order methods that we call *natural first-order methods*. More precisely, we establish that there is no natural first-order method for (2) requiring $\mathcal{O}(n^{2-\alpha} \log(1/\epsilon))$ FOs for any $\alpha > 0$. We also establish that there is no natural first-order method for (2) requiring $\tilde{\mathcal{O}}\left(n/\epsilon^\alpha\right)$ FOs for any $\alpha < 1/4$. The latter lower bound implies that the $\tilde{\mathcal{O}}\left(n/\epsilon^{1/3}\right)$ FO complexity of CSVRG is close to being tight. Table 1 summarizes our results.

**Comparing the methods for $\Theta(1/n)$ accuracy** According to the level of accuracy $\epsilon > 0$ that (2) requires, different methods might be more efficient than others. We remark that for the accuracy regime of $\epsilon = \mathcal{O}(1/n)$ all previous first-order methods require $\mathcal{O}(n^2)$ FOs while CSVRG requires only $\tilde{\mathcal{O}}(n^{4/3})$ FOs!

We remark that the accuracy regime $\epsilon = \mathcal{O}(1/n)$ for (2) is of particular interest and has been the initial motivation of this work. The reason for the latter lies on the generalization bounds in the strongly convex case. More precisely, the statistical error of empirical risk minimization in the strongly convex case is $\Theta(1/n)$ Shalev-Shwartz et al. (2010). Thus the accuracy of the respective finite-sum problem should be selected as $\epsilon = \mathcal{O}(1/n)$ so as to match the unavoidable statistical error (see Bottou and Bousquet (2007) for the respective discussion). For accuracy $\epsilon = \Theta(1/n)$, a very

| $\epsilon$-accuracy at each stage $i \in [n]$ | |
|---|---|
| Method | Number of FOs |
| StochasticGradientDescent | $\mathcal{O}(\frac{1}{\mu} n/\epsilon)$ |
| SVRG/SAGA/SARAH | $\mathcal{O}\left(n^2 \log(1/\epsilon) + \frac{L}{\mu} \cdot n \log(1/\epsilon)\right)$ |
| Katyusha | $\mathcal{O}\left(n^2 \log(1/\epsilon) + \sqrt{\frac{L}{\mu}} \cdot n^{3/2} \log(1/\epsilon)\right)$ |
| SGD-sparse (naive baseline) | $\mathcal{O}\left(\frac{|\mathcal{D}|G^3}{\mu} 1/\epsilon^2\right)$ |
| CSVRG (this work) | $\mathcal{O}\left(\frac{L^{2/3}G^{2/3}}{\mu} \cdot (n \log n)/\epsilon^{1/3} + \frac{L^2 G}{\mu^{5/2}} \cdot \log n/\sqrt{\epsilon}\right)$ |
| Lower Bound (this work) | $\Omega\left(n^2 \log(1/\epsilon)\right)$ |
| Lower Bound (this work) | $\Omega\left(n/\epsilon^{1/4}\right)$ |

Table 1: Number of FOs for the strongly convex case for an $\epsilon$-accurate solution for each stage $i \in [n]$.

interesting open question is bridging the gap between the $\mathcal{O}(n^{4/3})$ FO complexity of CSVRG and the $\Omega(n^{5/4})$ FO complexity that our lower bound suggests (setting $\epsilon = 1/n$ and $\alpha = 1/4$).

**Our Techniques** As already discussed the naive use of a VR method at each stage $i \in [n]$ results in overall $\mathcal{O}(n^2 \log(1/\epsilon))$ FOs. A first approach in order to alleviate the latter phenomenon is to sparsely use such a method across the $n$ stages and in most intermediate stages compute $\hat{x}_i \in \mathcal{D}$ via $\mathcal{O}(1)$ FOs using a stochastic gradient estimator. However both the *sparsity level* for using a VR method as well as the stochastic gradient estimator are crucial design choices for establishing accuracy $\epsilon > 0$ at each stage $i \in [n]$.

Our first-order method (CSVRG) adopts a variance-reduction pipeline along the above lines. CSVRG (Algorithm 1) maintains a direction $\tilde{\nabla}_i$ that at stage $i \in [n]$ tries to approximate $\sum_{k=1}^{i} \nabla f_k(\hat{x}_{i-1})/i$. Since computing the latter direction at each stage $i \in [n]$ would lead to $\Omega(n^2)$ FOs, CSVRG

- Sets $\tilde{\nabla}_i \leftarrow \sum_{k=1}^{i} \nabla f_k(\hat{x}_i)/i$ only in a very sparse sub-sequence of stages ($i$ FOs)
- Sets $\tilde{\nabla}_i \leftarrow (1 - \frac{1}{i})\tilde{\nabla}_i + \frac{1}{i}\nabla f_i(\hat{x}_{\text{prev}})$ in most stages (1 FO)

In order to output an $\epsilon$-accurate point $\hat{x}_i \in \mathcal{D}$ at stage $i \in [n]$, we initialize an internal subroutine (Algorithm 2) to the previous output $\hat{x}_{i-1} \in \mathcal{D}$ and perform $T_i$ stochastic gradient descent steps using the following novel gradient estimator

$$\nabla_i^t \leftarrow \left(1 - \frac{1}{i}\right)\left(\nabla f_{u_t}(x_i^t) - \nabla f_{u_t}(\hat{x}_{\text{prev}}) + \tilde{\nabla}_{i-1}\right) + \frac{1}{i}\nabla f_i(x_i^t)$$

where index $u_t$ is selected uniformly at random in $[i-1]$ and prev is the latest stage $\ell \leq i - 1$ at which $\tilde{\nabla}_\ell = \sum_{k=1}^{\ell} \nabla f_k(\hat{x}_\ell)/\ell$. Notice that the estimator $\nabla_i^t$ requires only 3 FOs. Combining all the latter, we establish that by an appropriate parametrization on the sparsity of the sub-sequence at which $\tilde{\nabla}_\ell = \sum_{k=1}^{\ell} \nabla f_k(\hat{x}_\ell)/\ell$ and an appropriate selection of iteration $T_i$, our first-order method CSVRG requires overall $\mathcal{O}(n \log n/\epsilon^{1/3})$ FOs.

## 1.1 RELATED WORK

Apart from the close relation of our work with the long line of research on variance reduction methods (see also Appendix A for a detailed discussion), the motivation of our work also relates with the line of research in incremental learning, the goal of which is adapting a model to new information, data or tasks without forgetting the ones that it was trained on earlier. The phenomenon of losing sight of old information is called *catastrophic forgetting* Castro et al. (2018); Goodfellow et al. (2014); Kirkpatrick et al. (2017); McCloskey and Cohen (1989); Mermillod et al. (2013) and it is one of the main challenges of incremental learning. There have been three main empirical approaches to tackle catastrophic forgetting, regularization based Nguyen et al. (2017a), memory based Tulving (1985), architecture based Yoon et al. (2017), as well as combination of the above Sodhani et al. (2020).

## 2 PRELIMINARIES

In this section we introduce some basic definitions and notation. We denote with $\mathrm{Unif}(1, \ldots, n)$ the uniform distribution over $\{1, \ldots, n\}$ and $[n] := \{1, \ldots, n\}$.

**Definition 1** (Strong Convexity). *A differentiable function $f : \mathcal{D} \mapsto \mathbb{R}$ is $\mu$-strongly convex in $\mathcal{D}$ if and only if for all $x, y \in \mathcal{D}$*

$$f(x) \geq f(y) + \nabla f(y)^\top (x - y) + \frac{\mu}{2} \|x - y\|^2 \tag{3}$$

In Problem (2) we make the assumption that $\mathcal{D}$ is convex and compact. We denote with $|\mathcal{D}|$ the diameter of $\mathcal{D}$, $|\mathcal{D}| = \max_{x,y \in \mathcal{D}} \|x - y\|_2$. The compactness of $\mathcal{D}$ also provides us with the property that each $f_i : \mathcal{D} \mapsto \mathbb{R}$ is also $G$-Lipschitz and $L$-smooth. More precisely,

- $|f(x) - f(y)| \leq G \cdot \|x - y\|$ ($G$-Lipschtiz)
- $\|\nabla f(x) - \nabla f(y)\| \leq L \cdot \|x - y\|$ ($L$-smooth)

where $G = \max_{x \in \mathcal{D}} \|\nabla f(x)\|_2$ and $L = \max_{x \in \mathcal{D}} \|\nabla^2 f(x)\|_2$. To simplify notation we denote as $g_i(x)$ the *prefix-function* of stage $i \in [n]$,

$$g_i(x) := \sum_{j=1}^{i} f_j(x)/i \tag{4}$$

We denote with $x_i^\star \in \mathcal{D}$ the minimizer of the function $g_i(x)$, $x_i^\star := \arg\min_{x \in \mathcal{D}} g_i(x)$. We denote the projection of $x \in \mathbb{R}^d$ to the set $\mathcal{D}$ as $\Pi_\mathcal{D}(x) := \arg\min_{z \in \mathcal{D}} \|x - z\|_2$. For a set $S \subseteq \mathbb{R}^d$, we denote $\Pi_\mathcal{D}(S) := \{\Pi_\mathcal{D}(x) : \text{for all } x \in S\}$. Throughout the paper we assume that each function $f_i(\cdot)$ in continual finite-sum minimization (2) is $\mu$-strongly convex, $L$-smooth and $G$-Lipschitz.

## 3 OUR RESULTS

We first state our main result establishing the existence of a first-order method for continual finite-sum minimization (2) with $\tilde{\mathcal{O}}\left(n/\epsilon^{1/3} + \log n/\sqrt{\epsilon}\right)$ FO complexity.

**Theorem 1.** *There exists a first-order method,* CSVRG *(Algorithm 1), for continual finite-sum minimization (2) with $\mathcal{O}\left(\frac{L^{2/3}G^{2/3}}{\mu} \cdot \frac{n \log n}{\epsilon^{1/3}} + \frac{L^2 G}{\mu^{5/2}} \cdot \frac{\log n}{\sqrt{\epsilon}}\right)$ FO complexity.*

As already mentioned, our method admits $\tilde{\mathcal{O}}\left(n/\epsilon^{1/3} + 1/\sqrt{\epsilon}\right)$ first-order complexity, significantly improving upon the $\mathcal{O}(n/\epsilon)$ of SGD and the $\mathcal{O}\left(n^2 \log(1/\epsilon)\right)$ of VR methods. The description of our method (Algorithm 1) as well as the main steps for establishing Theorem 1 are presented in Section 4.

In Theorem 2 we present the formal guarantees of the naive baseline result that is a *sparse variant* of SGD for continual finite-sum minimization (2) with $\mathcal{O}(1/\epsilon^2)$ FO complexity.

**Theorem 2.** *There exists a first-order method,* SGD-sparse *(Algorithm 4), for continual finite-sum minimization (2) with $\mathcal{O}\left(\frac{G^3 |\mathcal{D}| \log n}{\mu \epsilon^2}\right)$ FO complexity.*

Algorithm 4 as well as the proof of Theorem 2 are presented in Appendix I.

**Lower Bounds for Natural First-Order Methods** In view of the above, the question that naturally arises is whether there exists a method for continual finite-sum minimization (2) with $O(n^{2-\alpha} \log(1/\epsilon))$ FO complexity. Unfortunately the latter goal cannot be met for a wide class of first-order methods, that we call *natural*. The formal definition of *natural first-order method* is presented in Section 3.1. To this end, we remark that methods such that GD, SGD, SVRG, SAGA, SARAH, Katyusha and CSVRG (Algorithm 1) lie in the above class.

**Theorem 3.** *For any $\alpha > 0$, there is no natural first-order method for Problem (2) with $\mathcal{O}\left(n^{2-\alpha} \log(1/\epsilon)\right)$ FO complexity. Moreover for any $\alpha < 1/4$, there is no natural first-order method for continual finite-sum minimization (2) with $\mathcal{O}\left(n/\epsilon^\alpha\right)$ FO complexity.*

Due to space limitations the proof of Theorem 3 is presented in Appendix H.

**Remark 2.** *The $\tilde{\mathcal{O}}(n/\epsilon^{1/3} + 1/\sqrt{\epsilon})$ FO complexity of CSVRG is close to the $\Omega(n/\epsilon^{1/4})$ lower bound.*

### 3.1 NATURAL FIRST-ORDER METHODS

In this section we provide the formal definition of natural first-order method for which our lower bound stated in Theorem 3 applies. Before doing so we present the some definitions characterizing general first-order methods.

At an intuitive level a *first-order method* for continual finite-sum minimization (2) determines $\hat{x}_i \in \mathcal{D}$ by only requiring first-order access to the functions $f_1, \ldots, f_i$. Specifically at each stage $i \in [n]$, a first-order method makes queries of the form $\{\nabla f_j(x) \text{ for } j \leq i\}$ and uses this information to determine $\hat{x}_i \in \mathcal{D}$. The latter intuitive description is formalized in Definition 2.

**Definition 2.** *Let $\mathcal{A}$ be a first-order method for continual finite-sum minimization (2). At each stage $i \in [n]$,*

- *$\hat{x}_0 \in \mathcal{D}$ denotes the initial point of $\mathcal{A}$ and $\hat{x}_i \in \mathcal{D}$ denotes the output of $\mathcal{A}$ at stage $i \in [n]$.*

- *$x_i^t \in \mathcal{D}$ denotes the intermediate point of $\mathcal{A}$ at round $t \geq 1$ of stage $i \in [n]$ and $T_i \in \mathbb{N}$ the number of iterations during stage $i \in [n]$.*

- *At each round $t \in [T_i]$ of stage $i \in [n]$, $\mathcal{A}$ performs a set of first-order oracles denotes $Q_i^t$.*

    - *Each first-order oracle $q \in Q_i^t$ admits the form $q = (q_{\text{value}}, q_{\text{index}})$ where $q_{\text{value}} \in \mathcal{D}$ denotes the queried point and $q_{\text{index}} \leq i$ denotes the index of the queried function.*
    - *$\mathcal{A}$ computes the gradients $\{\nabla f_{q_{\text{index}}}(q_{\text{value}}) \text{ for all } q \in Q_i^t\}$ and uses this information to determine $x_i^{t+1} \in \mathcal{D}$.*

*The FO complexity of $\mathcal{A}$ is thus $\sum_{i \in [n]} \sum_{t=1}^{T_i} |Q_i^t|$.*

**Example 1.** *Gradient Descent at stage $i \in [n]$, sets $x_i^0 \leftarrow \hat{x}_{i-1}$ and for each round $t \in [T_i]$ peforms $x_i^t \leftarrow \Pi_{\mathcal{D}} \left[ x_i^{t-1} - \gamma \sum_{j=i}^i f_j(x_i^{t-1})/i \right]$. Finally it sets $\hat{x}_i \leftarrow x_i^{T_i}$. Thus in terms of Definition 2 the first-order oracles of GD at round $t \in T_i$ are $Q_i^t = \{ \left( x_i^{t-1}, 1 \right), \ldots, \left( x_i^{t-1}, i \right) \}$.*

In Definition 3 we formally define the notion of a *natural* first-order method. In simple terms a first-order method is called *natural* if during each stage $i \in [n]$ it only computes FOs of the form $\nabla f_j(x)$ where $x \in \mathcal{D}$ is previously generated point and $j \leq i$.

**Definition 3.** *A first-order method $\mathcal{A}$ is called natural iff at each step $t \in [T_i]$ of stage $i \in [n]$,*

- *For any query $q \in Q_i^t$, $q_{\text{index}} \leq i$ and $q_{\text{value}} \in \text{PP}_i^{t-1}$ where $\text{PP}_i^{t-1} := \left( \cup_{m \leq i-1} \cup_{\tau \leq T_m} x_m^\tau \right) \cup \left( \cup_{\tau \leq t-1} x_i^\tau \right) \cup \hat{x}_0$ ($\text{PP}_i^{t-1}$ stands for previous points)*

- *$x_i^t \in \Pi_{\mathcal{D}}(S)$ where $S$ is the linear span of $\left( \cup_{q \in Q_i^t} \nabla f_{q_{\text{index}}}(q_{\text{value}}) \right) \cup \text{PP}_i^{t-1}$.*

- *$\hat{x}_i \in \Pi_{\mathcal{D}}(S)$ where $S$ is the linear span of $\text{PP}_i^{T_i}$.*

**Remark 3.** *Definition 3 also captures randomized first-order methods such as SGD by considering $Q_i^t$ being a random set.*

Natural first-order methods is the straightforward extension of *linear-span methods* Bubeck (2015); Nesterov (2014) in the context of continual finite-sum minimization (2). Linear-span methods require that for any first-order oracle $\nabla f(x)$ computed at iteration $t$, $x \in \{x_0, x_1, \ldots, x_{t-1}\}$. Moreover $x_t$ is required to lie in the linear span of the previous points and the computed gradients. Linear-span methods capture all natural first-order optimization methods and have been used to establish the well-known $\Omega(1/\sqrt{\epsilon})$ and $\Omega\left( \sqrt{L/\mu} \log(1/\epsilon) \right)$ respectively for the convex and strongly convex case (see Section 3.5 in Bubeck (2015)).

## 4 CSVRG AND CONVERGENCE RESULTS

In this section we present our first-order method CSVRG that is able to achieve the guarantees established in Theorem 1. CSVRG is formally described in Algorithm 1.

---

**Algorithm 1** CSVRG

---

1: $\hat{x}_0 \in \mathcal{D}$, prev $\leftarrow 0$, $update \leftarrow false$
2: $\hat{x}_1 \leftarrow \text{GradientDescent}(\hat{x}_0)$, $\tilde{\nabla}_1 \leftarrow \nabla f_1(\hat{x}_1)$
3: **for** each stage $i = 2, \ldots, n$ **do**
4:     **if** $i - \text{prev} \geq \alpha \cdot i$ **then**
5:         $\tilde{\nabla}_{i-1} \leftarrow \frac{1}{i-1} \sum_{j=1}^{i-1} \nabla f_j(\hat{x}_{i-1})$             ▷ Compute $\nabla g_{i-1}(\hat{x}_{i-1})$ with $i-1$ FOs
6:         prev $\leftarrow i - 1$
7:         $update \leftarrow true$
8:     **end if**
9:     $T_i \leftarrow \mathcal{O}\left(\frac{L^2 G}{\mu^{5/2} i \sqrt{\epsilon}} + \frac{L^2 G^2 \alpha^2}{\mu \epsilon} + \frac{L^2}{\mu^2}\right)$ ▷ Number of iterations of Algorithm 2 at stage $i \in [n]$
10:     $\hat{x}_i \leftarrow \text{FUM}(\text{prev}, \tilde{\nabla}_{i-1}, T_i)$             ▷ Selection of $\hat{x}_i \in \mathcal{D}$ by Algorithm 2
11:     **if** $update$ **then**
12:         $\tilde{\nabla}_i \leftarrow \frac{1}{i} \sum_{j=1}^{i} \nabla f_j(\hat{x}_i)$               ▷ Compute $\nabla g_i(\hat{x}_i)$ with $i$ FOs
13:         prev $\leftarrow i$
14:         $update \leftarrow false$
15:     **else**
16:         $\tilde{\nabla}_i \leftarrow \left(1 - \frac{1}{i}\right) \tilde{\nabla}_{i-1} + \frac{1}{i} \nabla f_i(\hat{x}_{\text{prev}})$       ▷ Update $\tilde{\nabla}_i$ with 1 FO
17:     **end if**
18: **end for**

---

**Algorithm 2** **F**requent **U**pdate **M**ethod (FUM)

---

1: $\beta \leftarrow \frac{72 L^2}{\mu^2}$, $\mathcal{Z} \leftarrow \frac{T_i(T_i - 1)}{2} + (T_i + 1)(\beta - 1)$
2: $x_i^0 \leftarrow \hat{x}_{i-1}$                   ▷ Initialization at $\hat{x}_{i-1} \in \mathcal{D}$
3: **for** each round $t := 1, \ldots, T_i$ **do**
4:     Select $u_t \sim \text{Unif}(1, \ldots, i-1)$
5:     $\nabla_i^t \leftarrow \left(1 - \frac{1}{i}\right)\left(\nabla f_{u_t}(x_i^t) - \nabla f_{u_t}(\hat{x}_{\text{prev}}) + \tilde{\nabla}_{i-1}\right) + \frac{1}{i} \nabla f_i(x_i^t)$       ▷ 3 FOs
6:     $\gamma_t \leftarrow 4/(\mu(t + \beta))$              ▷ Step-size selection
7:     $x_i^{t+1} \leftarrow \Pi_{\mathcal{D}}\left(x_i^t - \gamma_t \nabla_i^t\right)$            ▷ Update $x_i^t \in \mathcal{D}$
8: **end for**
9: **Output:**  $\hat{x}_i \leftarrow \frac{1}{\mathcal{Z}} \sum_{s=0}^{T_i - 1} (s + \beta - 1) x_i^{t+1}$         ▷ Final output

---

We first remark that the computation of the output $\hat{x}_i \in \mathcal{D}$ for each stage $i \in [n]$, is performed at Step 10 of Algorithm 1 by calling Algorithm 2. Then Algorithm 2 initializes $x_i^0 := \hat{x}_{i-1}$ and performs $T_i$ stochastic gradient steps using the estimator

$$\nabla_i^t \leftarrow \left(1 - \frac{1}{i}\right)\left(\nabla f_{u_t}(x_i^t) - \nabla f_{u_t}(\hat{x}_{\text{prev}}) + \tilde{\nabla}_{i-1}\right) + \frac{1}{i}\nabla f_i(x_i^t)$$

To this end we remark that in order to compute $\nabla_i^t$ Algorithm 2 requires just 3 FOs. Before explaining the specific selection of the above estimator we start by explaining the role of $\tilde{\nabla}_i$.

**The role of $\tilde{\nabla}_i$:** In order to keep the variance of the estimator $\nabla_i^t$ low, we would ideally want $\tilde{\nabla}_{i-1} := \nabla g_{i-1}(\hat{x}_{i-1})$. The problem is that computing $\nabla g_i(\hat{x}_i)$ requires $i$ FOs meaning that performing such a computation at each stage $i \in [n]$, would directly result in $\Omega(n^2)$ FOs. In order to overcome the latter challenge, a full gradient $\nabla g_i(x)$ is computed very sparsely by Algorithm 1 across the $n$ stages. Specifically,

- In case $i - \text{prev} \geq \alpha \cdot i$ then $\tilde{\nabla}_i := \sum_{j=1}^{i} \nabla f_j(\hat{x}_i)/i$         ($i$ FOs)
- In case $i - \text{prev} < \alpha \cdot i$ then $\tilde{\nabla}_i := \left(1 - \frac{1}{i}\right) \tilde{\nabla}_{i-1} + \frac{1}{i} \nabla f_i(\hat{x}_{\text{prev}})$     (1 FO)

Thus the parameter $\alpha > 0$ controls the number of times Algorithm 1 reaches Steps 5 and 12 that at stage $i \in [n]$ require $i$ FOs. The latter is formally stated and established in Lemma 1.

**Lemma 1.** *Over a sequence of $n$ stages, Algorithm 1 reaches Step 5 and 12, $\lceil \log n/\alpha \rceil$ times.*

Combining the latter with the fact that Algorithm 2 requires $3T_i$ FOs at each stage $i \in [n]$ and the fact that Step 5 and 12 require at most $n$ FOs, we get that

**Corollary 1.** *Over a sequence of $n$ stages, Algorithm 1 requires $3\sum_{i=1}^{n} T_i + 2n\lceil \log n/\alpha \rceil$ FOs.*

Up next we present Theorem 4 establishing that for a specific selection of parameter $\alpha > 0$, Algorithm 1 guarantees that each $\hat{x}_i \in \mathcal{D}$ is an $\epsilon$-optimal point for the function $g_i(x)$.

**Theorem 4.** *Let a convex and compact set $\mathcal{D}$ and a sequence $\mu$-strongly convex functions $f_1, \ldots, f_n$ with $f_i : \mathcal{D} \mapsto \mathbb{R}$. Then Algorithm 1, with $T_i = 720GL^2/(\mu^{5/2}i\sqrt{\epsilon}) + 9L^{2/3}G^{2/3}/(\epsilon^{1/3}\mu) + 864L^2/\mu^2$ and $\alpha = \mu\epsilon^{1/3}/(20G^{2/3}L^{2/3})$, guarantees*

$$\mathbb{E}\left[g_i(\hat{x}_i)\right] - g_i(x_i^\star) \leq \epsilon \quad \text{for each stage } i \in [n]$$

*where $\hat{x}_i \in \mathcal{D}$ is the output of Algorithm 2 at Step 9 of Algorithm 1.*

The proof of Theorem 1 directly follows by Theorem 4 and Corollary 1. For completeness the proof Theorem 1 is presented in Appendix G. In the rest of the section we present the key steps for proving Theorem 4 (see Appendix F.2) for the full proof.

We first explain the role of the gradient estimator $\nabla_i^t$ in Step 5 of Algorithm 2. This estimator $\nabla_i^t$ of Step 5 in Algorithm 2 may seem unintuitive at first sight since it subtracts the term $\nabla f_{u_t}(\hat{x}_{\text{prev}})$. Interestingly enough the latter estimator-selection permits us to establish the following two key properties for $\nabla_i^t$.

**Lemma 2** (Unbiased). *Let $\nabla_i^t$ the gradient estimator used in Step 5 of Algorithm 2. Then for all $t \in [T_i]$, $\mathbb{E}\left[\nabla_i^t\right] = \nabla g_i(x_i^t)$.*

**Lemma 3** (Bounded Variance). *Let $\nabla_i^t$ the gradient estimator used in Step 5 of Algorithm 2. Then for all $t \in [T_i]$,*

$$\mathbb{E}\left[\left\|\nabla_i^t - \nabla g_i(x_t)\right\|_2^2\right] \leq 8L^2\mathbb{E}\left[\left\|x_i^t - x_i^\star\right\|_2^2\right] + 64\frac{L^2G^2}{\mu^2}\cdot\alpha^2 + 16L^2\mathbb{E}\left[\left\|x_{prev}^\star - \hat{x}_{prev}\right\|_2^2\right]$$

*where $\alpha > 0$ is the parameter used at Step 4 of Algorithm 1.*

The proof of Lemma 2 and Lemma 3 are respectively presented in Appendix C and Appendix D and consist one of the main technical contributions of this work. In Section 5 we explain why the specific estimator-selection $\nabla_i^t$ is crucial for establishing both Lemma 2 and 3.

In the rest of the section, we provide the main steps for establishing Theorem 4. Let us first inductively assume that $\mathbb{E}\left[g_j(\hat{x}_j) - g_j(x_j^\star)\right] \leq \epsilon$ for all $j \leq i - 1$. We use the latter to establish that $\mathbb{E}\left[g_i(\hat{x}_i) - g_i(x_i^\star)\right] \leq \epsilon$.

Notice that Step 4 of Algorithm 1 guarantees that prev $\leq i - 1$ at Step 10. Hence the induction hypothesis ensures that $\mathbb{E}\left[g_{\text{prev}}(\hat{x}_{\text{prev}}) - g_{\text{prev}}(x_{\text{prev}}^\star)\right] \leq \epsilon$ and thus by strong convexity $\mathbb{E}\left[\left\|\hat{x}_{\text{prev}} - x_{\text{prev}}^\star\right\|^2\right] \leq 2\epsilon/\mu$. Thus the bound in the variance of Lemma 3,

$$\mathbb{E}\left[\left\|\nabla_i^t - \nabla g_i(x_t)\right\|_2^2\right] \leq \mathcal{O}\left(L^2\mathbb{E}\left[\left\|x_i^t - x_i^\star\right\|_2^2\right] + \frac{L^2G^2}{\mu^2}\cdot\alpha^2 + \frac{L^2}{\mu}\epsilon\right) \tag{5}$$

Using the fact that $\mathbb{E}\left[\nabla_i^t\right] = \nabla g_i(x_i^t)$ and the bound in the variance $\mathbb{E}\left[\left\|\nabla_i^t - \nabla g_i(x_t)\right\|_2^2\right]$ of Equation 5, in Lemma 8 of Appendix E we establish that

$$\mathbb{E}\left[g_i(\hat{x}_i)\right] - g_i(x_i^\star) := \mathbb{E}\left[g_i(x_i^{T_i})\right] - g_i(x_i^\star) \leq \mathcal{O}\left(\frac{L^4}{\mu^3}\frac{\left\|x_i^0 - x_i^\star\right\|_2^2}{T_i^2} + \frac{L^2G^2}{\mu^3}\frac{\alpha^2}{T_i} + \frac{L^2\epsilon}{\mu^2T_i}\right).$$

Notice that at Step 2 of Algorithm 2, $x_i^0 := \hat{x}_{i-1} \in \mathcal{D}$ meaning that

$$
\begin{aligned}
\mathbb{E}\left[g_i(\hat{x}_i)\right] - g_i(x_i^\star) &\leq \mathcal{O}\left(\frac{L^4}{\mu^3}\frac{\left\|\hat{x}_{i-1} - x_i^\star\right\|_2^2}{T_i^2} + \frac{L^2G^2}{\mu^3}\frac{\alpha^2}{T_i} + \frac{L^2\epsilon}{\mu^2T_i}\right) \\
&\leq \mathcal{O}\left(\frac{L^4G^2}{\mu^5i^2\cdot T_i^2} + \frac{L^4\epsilon}{\mu^4T_i^2} + \frac{L^2G^2\alpha^2}{\mu^3T_i} + \frac{L^2\epsilon}{\mu^2T_i}\right)
\end{aligned}
$$

where the last inequality follows by $\|\hat{x}_{i-1} - x_i^\star\|_2^2 \leq 2\|\hat{x}_{i-1} - x_{i-1}^\star\|_2^2 + 2\|x_{i-1}^\star - x_i^\star\|_2^2$, $\|x_i^\star - x_{i-1}^\star\| \leq \mathcal{O}(G^2/(\mu i))$ and $\|\hat{x}_{i-1} - x_{i-1}^\star\| \leq \epsilon/\mu$. As a result, by taking $T_i = \mathcal{O}\left(\frac{L^2 G}{\mu^{5/2} i \sqrt{\epsilon}} + \frac{L^2 G^2 \alpha^2}{\mu^3 \epsilon} + \frac{L^2}{\mu^2}\right)$ we get that $\mathbb{E}\left[g_i(\hat{x}_i)\right] - g_i(x_i^\star) \leq \epsilon$.

## 5    ANALYZING THE ESTIMATOR $\nabla_i^t$

In this section we provide the key steps for proving Lemma 2 and Lemma 3 respectively.

**Unbias estimator:** The basic step for showing that $\nabla_i^t$ of Step 5 in Algorithm 2 is an unbiased estimator, $\mathbb{E}\left[\nabla_i^t\right] = \nabla g_i(x_i^t)$, is establishing the following property for $\tilde{\nabla}_{i-1}$ defined in Algorithm 1.

**Lemma 4.** *At Step 10 of Algorithm 1, it holds that $\tilde{\nabla}_{i-1} = \sum_{k=1}^{i-1} \nabla f_k(\hat{x}_{prev})/(i-1)$*

The proof of Lemma 4 is presented in Appendix C and admits an inductive proof on the steps of Algorithm 1. Once Lemma 4 is established the fact that $\mathbb{E}\left[\nabla_i^t\right] = \nabla f(x_i^t)$ easily follows by the selection of the estimator (see Step 5 of Algorithm 2). For the full proof of Lemma 2 we refer the reader to Appendix C.

**Bounding the Variance of $\nabla_i^t$:** The first step towards establishing Lemma 3 is presented in Lemma 5 providing a first bound on the variance $\mathbb{E}\left[\|\nabla_i^t - \nabla g_i(x_i^t)\|_2^2\right]$. The proof of Lemma 5 uses ideas derived from the analysis of VR methods and is presented in Appendix D.

**Lemma 5.** *For all rounds $t \in T_i$ of stage $i \in [n]$,*

$$\mathbb{E}\left[\|\nabla_i^t - \nabla g_i(x_i^t)\|_2^2\right] \leq 8L^2 \mathbb{E}\left[\|x_i^t - x_i^\star\|_2^2\right] + 8L^2 \mathbb{E}\left[\|x_i^\star - \hat{x}_{prev}\|_2^2\right] \tag{6}$$

*where prev is defined in Algorithm 1.*

The next step for establishing Lemma 3 is handling the term $\mathbb{E}\left[\|x_i^\star - \hat{x}_{\text{prev}}\|_2^2\right]$ of Equation 6. In order to do the latter in we exploit the strong convexity assumption so as to establish Lemma 6 the proof of which lies in Appendix B.

**Lemma 6.** *Let $\hat{x}_j \in \mathcal{D}$ the output of Algorithm 1 for stage $j \in [n]$. Then for all $i \in \{j+1, \ldots, n\}$,*

$$\|\hat{x}_j - x_i^\star\|_2^2 \leq \frac{8}{\mu^2}\left(\frac{G(i-j)}{i+j}\right)^2 + 2\|\hat{x}_j - x_j^\star\|_2^2$$

Now by applying Lemma 6 for $j := \text{prev}$ we get that

$$\begin{aligned}
\|x_i^\star - \hat{x}_{\text{prev}}\|_2^2 &\leq \frac{8}{\mu^2}\left(\frac{G(i-\text{prev})}{2\text{prev} + (i-\text{prev})}\right)^2 + 2\|\hat{x}_{\text{prev}} - x_{\text{prev}}^\star\|^2 \\
&\leq \frac{8}{\mu^2}\left(\frac{G(i-\text{prev})}{i}\right)^2 + 2\|\hat{x}_{\text{prev}} - x_{\text{prev}}^\star\|^2
\end{aligned}$$

Step 4 and 6 of Algorithm 1 ensure that at Step 10 of Algorithm 1 (when Algorithm 2 is called), $i - \text{prev} \leq \alpha \cdot i$. Thus,

$$\|x_i^\star - \hat{x}_{\text{prev}}\|_2^2 \leq \frac{8}{\mu^2}\left(\frac{G(i-\text{prev})}{i}\right)^2 + 2\|\hat{x}_{\text{prev}} - x_{\text{prev}}^\star\|^2 \leq \frac{32G^2}{\mu^2}\alpha^2 + 2\|\hat{x}_{\text{prev}} - x_{\text{prev}}^\star\|^2$$

and thus $\mathbb{E}\left[\|x_i^\star - \hat{x}_{\text{prev}}\|_2^2\right] \leq \frac{32G^2}{\mu^2}\alpha^2 + 2\mathbb{E}\left[\|\hat{x}_{\text{prev}} - x_{\text{prev}}^\star\|^2\right]$. Combining the latter with Equation 6 we overall get,

$$\mathbb{E}\left[\|\nabla_i^t - \nabla g_i(x_i^t)\|_2^2\right] \leq \mathcal{O}\left(L^2 \mathbb{E}\left[\|x_i^t - x_i^\star\|_2^2\right] + \frac{L^2 G^2}{\mu^2}\alpha^2 + L^2 \mathbb{E}\left[\|\hat{x}_{\text{prev}} - x_{\text{prev}}^\star\|^2\right]\right)$$

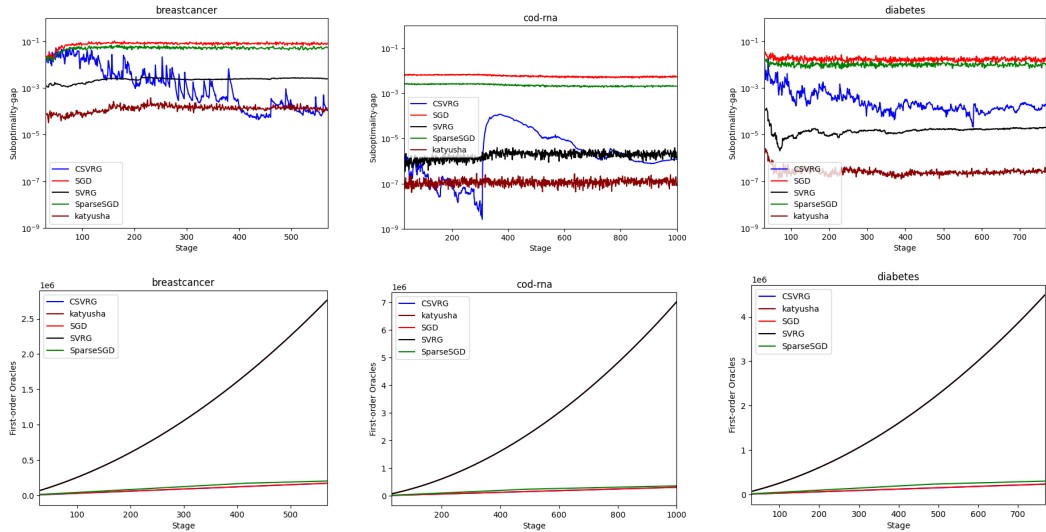

Table 2: Optimality gap as the stages progress on a ridge regression problem (averaged over 10 independent runs). CSVRG performs the exact same number of FOs with SGD and slightly less than SGD-sparse. Katyusha and SVRG perform the exact same number of FOs. CSVRG/SGD/SGD-sparse perform roughly 4% of the FOs of Katyusha/SVRG.

## 6 EXPERIMENTS

We experimentally evaluate the methods (SGD, SGD-sparse, Katyusha, SVRG and CSVRG) on a ridge regression task. Given some dataset $(a_i, b_i)_{i=1}^n \in \mathbb{R}^d \times \mathbb{R}$, at each stage $i \in [n]$ we consider the the finite sum objective $g_i(x) := \sum_{j=1}^i (a_j^\top x - b_j)^2/i + \lambda \|x\|_2^2$ with $\lambda = 10^{-3}$. We choose the latter setting so as to be able to compute the exact optimal solution at each stage $i \in [n]$.

For our experiments we use the datasets found in the LIBSVM packageChang and Lin (2011) for which we report our results. At each stage $i \in [n]$, we reveal a new data point $(a_i, b_i)$. In all of our experiments we run CSVRG with $\alpha = 0.3$ and $T_i = 100$. The inner iterations of SGD and SGD-sparse are appropriately selected so that their overall FO calls match the FO calls of CSVRG. At each stage $i \in [n]$ we run Katyusha Allen-Zhu (2017) and SVRG Johnson and Zhang (2013) on the prefix-sum function $g_i(x)$ with 10 outer iterations and 100 inner iterations.

In Table 2 we present the error achieved by each method at each stage $i \in [n]$ and the overall number of FOs until the respective stage. As our experimental evaluations reveal, CSVRG nicely interpolates between SGD/SGD-sparse and Katyusha/SVRG. More precisely, CSVRG achieves a significantly smaller suboptimality gap than SGD/SGD-sparse with the same number of FOs while it achieves a comparable one with Katyusha/SVRG with way fewer FOs. In Appendix K.2 we present further experimental evaluations as well as the exact parameters used for each method.

## 7 CONCLUSIONS

In this work we introduce *continual finite-sum optimization*, a tweak of standard finite-sum minimization, that given a sequence of functions asks for a sequence of $\epsilon$-accurate solutions for the respective prefix-sum function. For the strongly convex case we propose a first-order method with $\tilde{\mathcal{O}}(n/\epsilon^{1/3})$ FO complexity. This significantly improves upon the $\mathcal{O}(n/\epsilon)$ FOs that StochasticGradientDescent requires and upon the $\mathcal{O}(n^2 \log(1/\epsilon))$ FOs that VR methods require. We additionally prove that no *natural method* can achieve $\mathcal{O}\left(n^{2-\alpha} \log(1/\epsilon)\right)$ FO complexity for any $\alpha > 0$ and $\mathcal{O}\left(n/\epsilon^\alpha\right)$ FO complexity for $\alpha < 1/4$.

## 8 ACKNOWLEDGEMENTS

Authors acknowledge the constructive feedback of reviewers and the work of ICLR'24 program and area chairs. This work was supported by Hasler Foundation Program: Hasler Responsible AI (project number 21043), by the Swiss National Science Foundation (SNSF) under grant number 200021_20501, by the Army Research Office (Grant Number W911NF-24-1-0048) and by Innosuisse (contract agreement 100.960 IP-ICT).

**Limitations:** The theoretical guarantees of our proposed method are restricted to the strongly convex case. Extending our results beyond the strong convexity assumptions is a very promising research direction.

**Reproducibility Statement** In the appendix we have included the formal proofs of all the theorems and lemmas provided in the main part of the paper. In order to facilitate the reproducibility of our theoretical results, for each theorem we have created a separate self-contained section presenting its proof. Concerning the experimental evaluations of our work, we provide the code used in our experiments as well as the selected parameters in each of the presented methods.

**Ethics Statement** The authors acknowledge that they have read and adhere to the ICLR Code of Ethics.

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

## A    FURTHER RELATED WORK

In this section we present more extensively all work related to ours. This work lies in the intersection of convex finite sum minimization and incremental learning.

**Finite Sum Minimization:** As already mentioned, our work is closely related with the long line of research on variance reduction methods for strongly convex Nguyen et al. (2017b); Defazio et al. (2014); Allen-Zhu (2018b); Allen-Zhu (2017); Roux et al. (2012b); Xiao and Zhang (2014); Mokhtari et al. (2017); Johnson and Zhang (2013); Allen-Zhu (2017); Allen-Zhu (2018b); Zhang et al. (2013); Shalev-Shwartz and Zhang (2013a); Nguyen et al. (2017b); Defazio et al. (2014); Allen-Zhu (2018b); Allen-Zhu (2017); Roux et al. (2012b); Xiao and Zhang (2014), convex Lan et al. (2019b); Dubois-Taine et al. (2022); Song et al. (2020a); Lan et al. (2019a); Song et al. (2020b); Lin et al. (2018;?); Delyon and Juditsky (1993); Lin et al. (2015); Shalev-Shwartz and Zhang (2013b); Dubois-Taine et al. (2022) and non-convex Allen-Zhu and Yuan (2016); Allen-Zhu (2018a); Wang et al. (2019); Pham et al. (2020); Li and Richtárik (2021); Li et al. (2021); Fang et al. (2018a); Zhou et al. (2018); Reddi et al. (2016); Li and Li (2018); Kavis et al. (2022) finite-sum minimization (see also Appendix A for a more detailed discussion). As already mentioned the basic difference with our work comes from the fact these work concern the standard finite-sum setting while ours concerns its continual counter-part.

Our work leverages two components of variance reduction techniques. Recent works in variance reduction such as the SVRG algorithm Johnson and Zhang (2013) and the Katyusha algorithm Allen-Zhu (2017); Allen-Zhu (2018b) primarily leverage a full gradient computation, which is essentially used as a substitute for the gradients that are not computed in this iteration, in order to reduce the variance of their gradient estimator. Older methods, which seek to accelerate SGD, such as the RPDG method Lan and Zhou (2018) and the SAG method Roux et al. (2012a), which is a randomized variant of the IAG method Blatt et al. (2007), use a stochastic average of the current gradient that it sampled and substitutes the rest of the gradients with ones computed from previous iterations. Our method seeks to utilize the first technique, but the complexity constraint does not allow for this tool. We mitigate this problem using a technique similar to the incremental gradient methods. There are also several other methods and techniques that relate to accelerated stochastic gradient methods, both in the convex setting Song et al. (2020b); Allen-Zhu and Yuan (2016); Lan et al. (2019b); Lin et al. (2018); Kesten (1958); Delyon and Juditsky (1993); Zhang et al. (2013); Lin et al. (2015); Shalev-Shwartz and Zhang (2013b;a); Dubois-Taine et al. (2022) as well as in the non-convex setting Zhang and Xiao (2019); Fang et al. (2018b). In the past there have also been several works that focus on finite sum minimization in dynamic settings using SGD Kowshik et al. (2021b;a); Nagaraj et al. (2020)

**Incremental Learning:** The other side of this work is incremental learning, the goal of which is adapting a model to new information, data or tasks without forgetting the ones that it was trained on earlier. The phenomenon of losing sight of old information is called *catastrophic forgetting* Castro et al. (2018); Goodfellow et al. (2014); Kirkpatrick et al. (2017); McCloskey and Cohen (1989) and it is one of the main challenges of incremental learning. The area closest to our work from the field of incremental learning is lifelong learning. In this area catastrophic forgetting is expressed in the "stability-plasticity dilemma" Mermillod et al. (2013). There have been three main approaches to tackle this issue, regularization based approaches Nguyen et al. (2017a), memory based approaches Tulving (1985) and architecture based approaches Yoon et al. (2017); Lopez-Paz and Ranzato (2017). As shown in the current literature, combinations of these methods can be utilized in practice Sodhani et al. (2020) as well. All of the previous directions are studied in depth in Sodhani et al. (2022) and a brief overview of the techniques of the field is also presented in Castro et al. (2018). The area of lifelong learning also includes algorithms such as ELLA Ruvolo and Eaton (2013)

## B    PROOF OF LEMMA 6

In this section we provide the proof of Lemma 6.

**Lemma 6.** *Let $\hat{x}_j \in \mathcal{D}$ the output of Algorithm 1 for stage $j \in [n]$. Then for all $i \in \{j+1, \ldots, n\}$,*

$$\|\hat{x}_j - x_i^\star\|_2^2 \leq \frac{8}{\mu^2} \left( \frac{G(i-j)}{i+j} \right)^2 + 2 \|\hat{x}_j - x_j^\star\|_2^2$$

The proof of Lemma 6 follows by the proof of Lemma 7 that we state and prove up next. The proof of Lemma 6 is deferred at the end of this section.

**Lemma 7.** *For all $i \in [n-1]$ and $j \in [n-i]$,*

$$\left\|x_{i+j}^\star - x_i^\star\right\|_2 \leq \frac{2jG}{\mu(2i+j)}$$

*where $x_i^\star = \arg\min_{x \in \mathcal{D}} g_i(x)$ and $x_{i+j}^\star = \arg\min_{x \in \mathcal{D}} g_{i+j}(x)$.*

*Proof of Lemma 7.* By the optimality of $x_{i+j}^\star \in \mathcal{D}$ for the function $g_{i+j}(x)$ we get that

$$\langle \nabla g_{i+j}(x_{i+j}^\star), x_i^\star - x_{i+j}^\star \rangle \geq 0$$

The inequality is shown in (Boyd and Vandenberghe, 2004, equation 4.21).

From the strong convexity of $g_{i+j}(x)$ we get that

$$
\begin{aligned}
\frac{\mu}{2}\left\|x_{i+j}^\star - x_i^\star\right\|_2^2 &\leq g_{i+j}(x_i^\star) - g_{i+j}(x_{i+j}^\star) - \langle \nabla g_{i+j}(x_{i+j}^\star), x_i^\star - x_{i+j}^\star \rangle \\
&\leq g_{i+j}(x_i^\star) - g_{i+j}(x_{i+j}^\star) \\
&= \frac{1}{i+j}\sum_{k=1}^{i+j} f_k(x_i^\star) - \frac{1}{i+j}\sum_{k=1}^{i+j} f_k(x_{i+j}^\star) \\
&= \frac{i}{i+j}\left(g_i(x_i^\star) - g_i(x_{i+j}^\star)\right) + \frac{1}{i+j}\sum_{k=i+1}^{i+j}\left(f_k(x_i^\star) - f_k(x_{i+j}^\star)\right) \\
&\leq \frac{i}{i+j}\left(-\frac{\mu}{2}\left\|x_{i+j}^\star - x_i^\star\right\|_2^2 - \langle \nabla g_i(x_i^\star), x_{i+j}^\star - x_i^\star \rangle\right) \\
&\quad + \frac{1}{i+j}\sum_{k=i+1}^{i+j}\left(f_k(x_i^\star) - f_k(x_{i+j}^\star)\right) \\
&\leq \frac{i}{i+j}\left(-\frac{\mu}{2}\left\|x_{i+j}^\star - x_i^\star\right\|_2^2\right) + \frac{1}{i+j}\sum_{k=i+1}^{i+j} G\left\|x_{i+j}^\star - x_i^\star\right\|_2 \\
&\leq \frac{i}{i+j}\left(-\frac{\mu}{2}\left\|x_{i+j}^\star - x_i^\star\right\|_2^2\right) + \frac{jG\left\|x_{i+j}^\star - x_i^\star\right\|_2}{i+j}
\end{aligned}
$$

$\square$

We conclude the section with the proof of Lemma 6.

*Proof of Lemma 6.* By the inequality $\|a+b\|_2^2 \leq 2\|a\|_2^2 + 2\|b\|_2^2$ we get that

$$
\begin{aligned}
\left\|\hat{x}_j - x_i^\star\right\|_2^2 &\leq 2\left(\left\|x_i^\star - x_j^\star\right\|_2^2 + \left\|\hat{x}_j - x_j^\star\right\|_2^2\right) \\
&\leq \frac{8}{\mu^2}\left(\frac{Gj}{2i+j}\right)^2 + \left\|\hat{x}_j - x_j^\star\right\|_2^2
\end{aligned}
$$

The second inequality comes from Lemma 7 and strong convexity. $\square$

## C  PROOF OF LEMMA 2

In this section we provide the proof of Lemma 2.

**Lemma 2** (Unbiased). *Let $\nabla_i^t$ the gradient estimator used in Step 5 of Algorithm 2. Then for all $t \in [T_i]$, $\mathbb{E}\left[\nabla_i^t\right] = \nabla g_i(x_i^t)$.*

The basic step in order to establish Lemma 2 is Lemma 4 that we present up next. The proof of Lemma 2 is deferred at the end of the section.

**Lemma 4.** *At Step* 10 *of Algorithm 1, it holds that* $\tilde{\nabla}_{i-1} = \sum_{k=1}^{i-1} \nabla f_k(\hat{x}_{prev})/(i-1)$

*Proof of Lemma 4.* We will inductively establish Lemma 4. Notice that after stage $i = 1$, Algorithm 1 sets $\tilde{\nabla}_1 = \nabla f_1(\hat{x}_1)$. Up next we show that in case the induction hypothesis holds for stage $i - 1$ then it must essentially hold for stage $i$. Up next we consider the following 3 mutually exclusive cases:

1. $i - \text{prev} \geq \alpha i$ meaning that prev is updated in this stage.

2. $(i - 1) - \text{prev} \geq \alpha(i - 1)$ meaning that prev was update in the previous stage.

3. $i - \text{prev} < \alpha i$ and $(i - 1) - \text{prev} < \alpha(i - 1)$.

For the first case Algorithm 1 reaches Step 5 and thus $\tilde{\nabla}_{i-1} = \sum_{j=1}^{i-1} \nabla f_j(\hat{x}_{i-1})/i - 1$. At the same time prev is set to $i - 1$ meaning that prev $= i - 1$ and $\tilde{\nabla}_{i-1} = \frac{1}{i-1} \sum_{k=1}^{i-1} \nabla f_k(\hat{x}_{\text{prev}})$.

For the second case, Algorithm 1 had reached Step 12 at stage $i - 1$ meaning that $\tilde{\nabla}_{i-1} = \sum_{j=1}^{i-1} \nabla f_j(\hat{x}_{i-1})/i - 1$. At the same time prev was set to $i - 1$, meaning that prev $= i - 1$ and $\tilde{\nabla}_{i-1} = \frac{1}{i-1} \sum_{k=1}^{i-1} \nabla f_k(\hat{x}_{\text{prev}})$.

For the third case, from the inductive hypothesis we have that:

$$\tilde{\nabla}_{i-2} = \frac{1}{i-2} \sum_{k=1}^{i-2} \nabla f_k(\hat{x}_{\text{prev}})$$

At stage $i - 1$, $\tilde{\nabla}_{i-1}$ was calculated according to Step 16 of Algorithm 1. As a result,

$$\tilde{\nabla}_{i-1} = \left(1 - \frac{1}{i-1}\right) \tilde{\nabla}_{i-2} + \frac{1}{i-1} \nabla f_{i-1}(\hat{x}_{\text{prev}})$$

$$= \frac{i-2}{i-1} \frac{1}{i-2} \sum_{k=1}^{i-2} \nabla f_k(\hat{x}_{\text{prev}}) + \frac{1}{i-1} \nabla f_{i-1}(\hat{x}_{\text{prev}})$$

$$= \frac{1}{i-1} \sum_{k=1}^{i-1} \nabla f_k(\hat{x}_{\text{prev}})$$

$\square$

*Proof of Lemma 2.* Let $\mathcal{F}_t^i$ denote the filtration until step $t \in [T_i]$ of stage $i \in [n]$.

By the definition of $\nabla_i^t$ at Step 5 of Algorithm 2 we know that

$$
\begin{aligned}
\mathbb{E}\left[\nabla_i^t \mid \mathcal{F}_t^i\right] &= \left(1 - \frac{1}{i}\right)\left(\mathbb{E}\left[\nabla f_{u_t}(x_i^t) - \nabla f_{u_t}(\hat{x}_{\text{prev}}) \mid \mathcal{F}_t^i\right] + \tilde{\nabla}_{i-1}\right) + \frac{1}{i}\nabla f_i(x_i^t) \\
&= \frac{i-1}{i}\left(\sum_{k=1}^{i-1} \Pr\left[u_t = k\right]\left(\nabla f_k(x_i^t) - \nabla f_k(\hat{x}_{\text{prev}})\right) + \tilde{\nabla}_{i-1}\right) + \frac{1}{i}\nabla f_i(x_i^t) \\
&= \frac{i-1}{i}\left(\frac{1}{i-1} \sum_{k=1}^{i-1}\left(\nabla f_k(x_i^t) - \nabla f_k(\hat{x}_{\text{prev}})\right) + \tilde{\nabla}_{i-1}\right) + \frac{1}{i}\nabla f_i(x_i^t) \\
&= \frac{1}{i} \sum_{k=1}^{i-1} \nabla f_k(x_i^t) - \frac{1}{i} \sum_{k=1}^{i-1} \nabla f_k(\hat{x}_{\text{prev}}) + \frac{i-1}{i}\tilde{\nabla}_{i-1} + \frac{1}{i}\nabla f_i(x_i^t)
\end{aligned}
$$

Lemma 4 ensures that $\frac{i-1}{i}\tilde{\nabla}_{i-1} := \frac{1}{i}\sum_{k=1}^{i-1}\nabla f_k(\hat{x}_{\text{prev}})$ meaning that

$$\mathbb{E}\left[\nabla_i^t \mid \mathcal{F}_t^i\right] = \frac{1}{i}\sum_{k=1}^{i-1}\nabla f_k(x_i^t) + \frac{1}{i}\nabla f_i(x_i^t) = \nabla g_i(x_i^t).$$

$\square$

## D  PROOF OF LEMMA 3

In this section we provide the proof of Lemma 3.

**Lemma 3** (Bounded Variance). *Let $\nabla_i^t$ the gradient estimator used in Step $5$ of Algorithm 2. Then for all $t \in [T_i]$,*

$$\mathbb{E}\left[\left\|\nabla_i^t - \nabla g_i(x_t)\right\|_2^2\right] \le 8L^2\mathbb{E}\left[\left\|x_i^t - x_i^\star\right\|_2^2\right] + 64\frac{L^2G^2}{\mu^2}\cdot\alpha^2 + 16L^2\mathbb{E}\left[\left\|x_{prev}^\star - \hat{x}_{prev}\right\|_2^2\right]$$

*where $\alpha > 0$ is the parameter used at Step $4$ of Algorithm 1.*

In order to prove Lemma 3 we first state and establish Lemma 5

**Lemma 5.** *For all rounds $t \in T_i$ of stage $i \in [n]$,*

$$\mathbb{E}\left[\left\|\nabla_i^t - \nabla g_i(x_i^t)\right\|_2^2\right] \le 8L^2\mathbb{E}\left[\left\|x_i^t - x_i^\star\right\|_2^2\right] + 8L^2\mathbb{E}\left[\left\|x_i^\star - \hat{x}_{prev}\right\|_2^2\right] \tag{6}$$

*where prev is defined in Algorithm 1.*

*Proof of Lemma 5.* By substituting the definition of $\nabla_i^t$:

$$\mathbb{E}[\|\nabla_i^t - \nabla g_i(x_i^t)\|_2^2] = \left(1 - \frac{1}{i}\right)^2 \cdot \mathbb{E}\left[\left\|\nabla f_{u_t}(x_i^t) - \nabla f_{u_t}(\hat{x}_{\text{prev}}) + \tilde{\nabla}_{i-1} - \nabla g_{i-1}(x_i^t)\right\|_2^2\right]$$

$$\le 2\left(1 - \frac{1}{i}\right)^2 \cdot \mathbb{E}\left[\left\|\nabla f_{u_t}(x_i^t) - \nabla f_{u_t}(\hat{x}_{\text{prev}})\right\|_2^2\right] + 2\left(1 - \frac{1}{i}\right)^2 \cdot \mathbb{E}\left[\left\|\frac{1}{i-1}\sum_{k=1}^{i-1}\nabla f_k(x_i^t) - \tilde{\nabla}_{i-1}\right\|_2^2\right]$$

$$= 2\left(1 - \frac{1}{i}\right)^2 \cdot \mathbb{E}\left[\left\|\nabla f_{u_t}(x_i^t) - \nabla f_{u_t}(\hat{x}_{\text{prev}})\right\|_2^2\right]$$

$$+ 2\left(1 - \frac{1}{i}\right)^2 \cdot \mathbb{E}\left[\left\|\frac{1}{i-1}\sum_{k=1}^{i-1}(\nabla f_k(x_i^t) - \nabla f_k(\hat{x}_{\text{prev}}))\right\|_2^2\right]$$

$$\le 2\left(1 - \frac{1}{i}\right)^2 \cdot L^2 \cdot \mathbb{E}\left[\left\|x_i^t - \hat{x}_{\text{prev}}\right\|_2^2\right] + 2\left(1 - \frac{1}{i}\right)^2 \frac{1}{(i-1)^2}\mathbb{E}\left[\left\|\sum_{k=1}^{i-1}(\nabla f_k(x_i^t) - \nabla f_k(\hat{x}_{\text{prev}}))\right\|_2^2\right]$$

$$\le 2\left(1 - \frac{1}{i}\right)^2 \cdot L^2 \cdot \mathbb{E}\left[\left\|x_i^t - \hat{x}_{\text{prev}}\right\|_2^2\right] + 2\left(1 - \frac{1}{i}\right)^2 \frac{i-1}{(i-1)^2}\mathbb{E}\left[\sum_{k=1}^{i-1}\left\|\nabla f_k(x_i^t) - \nabla f_k(\hat{x}_{\text{prev}})\right\|_2^2\right]$$

$$= 2\left(1 - \frac{1}{i}\right)^2 \cdot L^2 \cdot \mathbb{E}\left[\left\|x_i^t - \hat{x}_{\text{prev}}\right\|_2^2\right] + 2\left(1 - \frac{1}{i}\right)^2 \frac{1}{(i-1)}\sum_{k=1}^{i-1}\mathbb{E}\left[\left\|\nabla f_k(x_i^t) - \nabla f_k(\hat{x}_{\text{prev}})\right\|_2^2\right]$$

$$= 2\left(1 - \frac{1}{i}\right)^2 \cdot L^2 \cdot \mathbb{E}\left[\left\|x_i^t - \hat{x}_{\text{prev}}\right\|_2^2\right] + 2\left(1 - \frac{1}{i}\right)^2 \frac{1}{(i-1)}\sum_{k=1}^{i-1}\mathbb{E}\left[\left\|\nabla f_k(x_i^t) - \nabla f_k(\hat{x}_{\text{prev}})\right\|_2^2\right]$$

$$\le 2\left(1 - \frac{1}{i}\right)^2 \cdot L^2 \cdot \mathbb{E}\left[\left\|x_i^t - \hat{x}_{\text{prev}}\right\|_2^2\right] + 2\left(1 - \frac{1}{i}\right)^2 \frac{L^2}{(i-1)}\sum_{k=1}^{i-1}\mathbb{E}\left[\left\|x_i^t - \hat{x}_{\text{prev}}\right\|_2^2\right]$$

$$= 4\left(1 - \frac{1}{i}\right)^2 L^2\mathbb{E}\left[\left\|x_i^t - \hat{x}_{\text{prev}}\right\|_2^2\right]$$

$$\le 8L^2\left(1 - \frac{1}{i}\right)^2\left(\mathbb{E}[\|x_i^t - x_i^\star\|_2^2] + \mathbb{E}[\|x_i^\star - \hat{x}_{\text{prev}}\|_2^2]\right)$$

$$\square$$

*Proof of Lemma 3.* Applying Lemma 6 for $j := \text{prev}$ we get that,

$$
\begin{aligned}
\mathbb{E}\left[\|x_i^\star - \hat{x}_{\text{prev}}\|_2^2\right] &\leq \frac{8}{\mu^2}\left(\frac{G(i-\text{prev})}{i+\text{prev}}\right)^2 + 2\mathbb{E}\left[\|x_{\text{prev}}^\star - \hat{x}_{\text{prev}}\|_2^2\right] \\
&\leq \frac{8}{\mu^2}\left(\frac{G(i-\text{prev})}{i}\right)^2 + 2\mathbb{E}\left[\|x_{\text{prev}}^\star - \hat{x}_{\text{prev}}\|_2^2\right] \\
&\leq \frac{8}{\mu^2}\left(\frac{Gi\alpha}{i}\right)^2 + 2\mathbb{E}\left[\|x_{\text{prev}}^\star - \hat{x}_{\text{prev}}\|_2^2\right] \\
&= \frac{8G^2}{\mu^2}\alpha^2 + 2\mathbb{E}\left[\|x_{\text{prev}}^\star - \hat{x}_{\text{prev}}\|_2^2\right]
\end{aligned}
$$

The third inequality comes from the fact that $i-\text{prev} \leq \alpha i$ which is enforced by Step 4 of Algorithm 1. By Lemma 5 we get that

$$
\begin{aligned}
\mathbb{E}[\|\nabla_i^t - \nabla g_i(x_i^t)\|_2^2] &\leq 8L^2\mathbb{E}[\|x_i^t - x_i^\star\|_2^2] + 8L^2\mathbb{E}[\|x_i^\star - \hat{x}_{\text{prev}}\|_2^2] \\
&\leq 8L^2\mathbb{E}[\|x_i^t - x_i^\star\|_2^2] + \frac{64G^2L^2}{\mu^2}\alpha^2 + 16L^2\mathbb{E}\left[\|x_{\text{prev}}^\star - \hat{x}_{\text{prev}}\|_2^2\right]
\end{aligned}
$$

which concludes the proof. $\square$

## E  PROOF OF LEMMA 8

In this section we provide proof for Lemma 8.

**Lemma 8** (Convergence Bound). *If $\mathbb{E}[g_j(\hat{x}_j)] - g_j(x_j^\star) \leq \epsilon$ for all stages $j \in [i-1]$ then*

$$
\mathbb{E}\left[g_i(\hat{x}_i) - g_i(x_i^\star)\right] \leq \frac{1}{\mathcal{Z}}\left(\frac{\mu}{8}(\beta-1)(\beta-2)\mathbb{E}\left[\|x_i^0 - x_i^\star\|_2^2\right] + \kappa^2 G^2\alpha^2\frac{1}{\mu}T_i + 2\kappa^2\epsilon T_i\right) \quad (7)
$$

*where $\mathcal{Z} = \frac{T_i(T_i-1)}{2} + (T_i+1)(\beta-1)$, $\kappa = 6L/\mu$ and $\beta = 72L^2/\mu^2$.*

*Proof of Lemma 8.* In order to establish Lemma 8 we use Lemma 9 which is similar to (Allen-Zhu, 2018b, Lemma 3.2). The proof of Lemma 9 is presented in Appendix E.1.

**Lemma 9.** *In case $\mathbb{E}[\nabla_i^t] = \nabla g_i(x_i^t)$ then*

$$
\mathbb{E}\left[g_i(x_i^{t+1}) - g_i(x_i^\star)\right] \leq \mathbb{E}\left[\frac{9}{16}\gamma_t\|\nabla_i^t - \nabla g_i(x_i^t)\|_2^2 + \frac{1-\mu\gamma_t}{2\gamma_t}\|x_i^\star - x_i^t\|_2^2 - \frac{1}{2\gamma_t}\|x_i^\star - x_i^{t+1}\|_2^2\right]
$$
$$(8)$$

*where $x_i^{t+1} := \Pi_{\mathcal{D}}(x_i^t - \gamma_t\nabla_i^t)$, $\gamma_t = \frac{4}{\mu(t+\beta)}$ and $\beta = 72L^2/\mu^2$ (see Step 1 of Algorithm 2).*

**Lemma 2** (Unbiased). *Let $\nabla_i^t$ the gradient estimator used in Step 5 of Algorithm 2. Then for all $t \in [T_i]$, $\mathbb{E}\left[\nabla_i^t\right] = \nabla g_i(x_i^t)$.*

Since Lemma 2 establishes that $\mathbb{E}[\nabla_i^t] = \nabla g_i(x_i^t)$ we can apply Lemma 9. In order to bound the variance $\mathbb{E}\left[\|\nabla_i^t - \nabla g_i(x_i^t)\|_2^2\right]$ appearing in the RHS of Equation 8, we use Lemma 3 (see Appendix D for its proof).

**Lemma 3** (Bounded Variance). *Let $\nabla_i^t$ the gradient estimator used in Step 5 of Algorithm 2. Then for all $t \in [T_i]$,*

$$
\mathbb{E}\left[\|\nabla_i^t - \nabla g_i(x_t)\|_2^2\right] \leq 8L^2\mathbb{E}\left[\|x_i^t - x_i^\star\|_2^2\right] + 64\frac{L^2G^2}{\mu^2}\cdot\alpha^2 + 16L^2\mathbb{E}\left[\|x_{prev}^\star - \hat{x}_{prev}\|_2^2\right]
$$

*where $\alpha > 0$ is the parameter used at Step 4 of Algorithm 1.*

As discussed above by combining Lemma 9 with Lemma 3, we get the following:

$$
\begin{aligned}
\mathbb{E}\left[g_i(x_i^{t+1}) - g_i(x_i^\star)\right] \leq\; & \frac{\gamma_t^{-1} + 9L^2\gamma_t - \mu}{2}\mathbb{E}\left[\left\|x_i^t - x_i^\star\right\|_2^2\right] - \frac{\gamma_t^{-1}}{2}\mathbb{E}\left[\left\|x_i^{t+1} - x_i^\star\right\|_2^2\right] \\
& + 36\frac{L^2 G^2}{\mu^2}\alpha^2\gamma_t + 9L^2\gamma_t\mathbb{E}\left[\left\|x_{\mathrm{prev}}^\star - \hat{x}_{\mathrm{prev}}\right\|_2^2\right]
\end{aligned}
\tag{9}
$$

By the strong-convexity of $g_{\mathrm{prev}}(\cdot)$ we get

$$
\left\|x_{\mathrm{prev}}^\star - \hat{x}_{\mathrm{prev}}\right\|_2^2 \leq \frac{2}{\mu}\left(g_{\mathrm{prev}}(\hat{x}_{\mathrm{prev}}) - g_{\mathrm{prev}}(x_{\mathrm{prev}}^\star)\right)
$$

and from our inductive hypothesis

$$
\mathbb{E}\left[g_{\mathrm{prev}}(\hat{x}_{\mathrm{prev}}) - g_{\mathrm{prev}}(x_{\mathrm{prev}}^\star)\right] \leq \epsilon
$$

Notice that by the selection of $\gamma_t = 4/(\mu(t + \beta))$ and $\beta = 72L^2/\mu^2$ of Algorithm 2 we get that

$$
9L^2\gamma_t = \frac{36L^2}{\mu(t + \beta)} \leq \frac{36L^2}{\mu\beta} \leq \frac{36L^2}{\mu 72L^2/\mu^2} \leq \frac{36L^2}{72L^2/\mu} = \frac{\mu}{2}
$$

Using the previous inequalities in Equation 9 we get that

$$
\begin{aligned}
\mathbb{E}\left[g_i(x_{t+1}) - g_i(x_i^\star)\right] \leq\; & \frac{\gamma_t^{-1} - \mu/2}{2}\mathbb{E}\left[\left\|x_i^t - x_i^\star\right\|_2^2\right] - \frac{\gamma_t^{-1}}{2}\mathbb{E}\left[\left\|x_i^{t+1} - x_i^\star\right\|_2^2\right] \\
& + \kappa^2 G^2\alpha^2\gamma_t + \frac{18L^2\gamma_t}{\mu}\epsilon
\end{aligned}
\tag{10}
$$

Since $\kappa = 6L/\mu$.

Multiplying both parts of Equation 10 with $(t + \beta - 1)$ and substituting $\gamma_t = 4/(\mu(t + \beta))$ we get that

$$
\begin{aligned}
(t + \beta - 1)\mathbb{E}\left[g_i(x_i^{t+1}) - g_i(x_i^\star)\right] \leq\; & \frac{\mu(t + \beta - 1)(t + \beta - 2)}{8}\mathbb{E}\left[\left\|x_i^t - x_i^\star\right\|_2^2\right] \\
& - \frac{\mu(t + \beta - 1)(t + \beta)}{8}\mathbb{E}\left[\left\|x_i^{t+1} - x_i^\star\right\|_2^2\right] \\
& + \left(\kappa^2 G^2\alpha^2 + \frac{18L^2}{\mu}\epsilon\right) \cdot \frac{4}{\mu(t + \beta)} \cdot (t + \beta - 1)
\end{aligned}
$$

where $\beta = 72L^2/\mu^2$. By setting $\kappa = 6L/\mu$ we get that,

$$
\begin{aligned}
(t + \beta - 1)\mathbb{E}\left[g_i(x_i^{t+1}) - g_i(x_i^\star)\right] \leq\; & \frac{\mu(t + \beta - 1)(t + \beta - 2)}{8}\mathbb{E}\left[\left\|x_i^t - x_i^\star\right\|_2^2\right] \\
& - \frac{\mu(t + \beta - 1)(t + \beta)}{8}\mathbb{E}\left[\left\|x_i^{t+1} - x_i^\star\right\|_2^2\right] \\
& + \left(\kappa^2\frac{G^2}{\mu}\alpha^2 + 2\kappa^2\epsilon\right)
\end{aligned}
$$

By taking the summation over all iterations $t = 0, \ldots, T_i - 1$, we get

$$
\begin{aligned}
\sum_{t=0}^{T_i-1}(t + \beta - 1)\mathbb{E}[g_i(x_i^{t+1}) - g_i(x_i^\star)] \leq\; & \frac{\mu}{8}\sum_{t=0}^{T_i-1}(t + \beta - 1)(t + \beta - 2)\mathbb{E}\left[\left\|x_i^t - x_i^\star\right\|_2^2\right] \\
& - \frac{\mu}{8}\sum_{t=0}^{T_i-1}(t + \beta - 1)(t + \beta)\mathbb{E}\left[\left\|x_i^{t+1} - x_i^\star\right\|_2^2\right] \\
& + T_i\left(\kappa^2\frac{G^2}{\mu}\alpha^2 + 2\kappa^2\epsilon\right)
\end{aligned}
$$

As a result, we get that

$$
\begin{aligned}
\sum_{t=0}^{T_i-1} (t + \beta - 1)\mathbb{E}[g_i(x_i^{t+1}) - g_i(x_i^\star)] \leq & \ \frac{\mu}{8}(\beta - 1)(\beta - 2)\mathbb{E}\left[\left\|x_i^0 - x_i^\star\right\|_2^2\right] \\
& - \ \frac{\mu}{8}(T_i + \beta - 2)(T_i + \beta - 1)\mathbb{E}\left[\left\|x_{T_i} - x_i^\star\right\|_2^2\right] \\
& + \ T_i\left(\kappa^2\frac{G^2}{\mu}\alpha^2 + 2\kappa^2\epsilon\right)
\end{aligned}
$$

Dividing by $\mathcal{Z} = \sum_{t=0}^{T_i-1}(t + \beta - 1) = T_i(T_i - 1)/2 + T_i(\beta - 1)$ and using the convexity of $g_i$ we get that

$$
\begin{aligned}
\mathbb{E}\left[g_i\left(\frac{1}{\mathcal{Z}}\sum_{t=0}^{T_i-1}(t + \beta - 1)x_i^{t+1}\right) - g_i(x_i^\star)\right] \leq & \ \frac{1}{\mathcal{Z}}\frac{\mu}{8}(\beta - 1)(\beta - 2)\mathbb{E}\left[\left\|x_i^0 - x_i^\star\right\|_2^2\right] + \kappa^2\frac{G^2}{\mu}\alpha^2 T_i\frac{1}{\mathcal{Z}} \\
& + \ 2\kappa^2\frac{T_i\epsilon}{\mathcal{Z}}
\end{aligned}
$$

As a result,

$$
\mathbb{E}\left[g_i\left(\hat{x}_i\right) - g_i(x_i^\star)\right] \leq \frac{1}{\mathcal{Z}}\left(\frac{\mu}{8}(\beta - 1)(\beta - 2)\mathbb{E}\left[\left\|x_i^0 - x_i^\star\right\|_2^2\right] + \kappa^2 G^2\alpha^2\frac{1}{\mu}T_i + 2\kappa^2\epsilon T_i\right)
$$

$\square$

### E.1 PROOF OF LEMMA 9

In this section we provide the proof for Lemma 9.

**Lemma 9.** *In case* $\mathbb{E}[\nabla_i^t] = \nabla g_i(x_i^t)$ *then*

$$
\mathbb{E}\left[g_i(x_i^{t+1}) - g_i(x_i^\star)\right] \leq \mathbb{E}\left[\frac{9}{16}\gamma_t\left\|\nabla_i^t - \nabla g_i(x_i^t)\right\|_2^2 + \frac{1 - \mu\gamma_t}{2\gamma_t}\left\|x_i^\star - x_i^t\right\|_2^2 - \frac{1}{2\gamma_t}\left\|x_i^\star - x_i^{t+1}\right\|_2^2\right]
$$

(8)

*where* $x_i^{t+1} := \Pi_{\mathcal{D}}\left(x_i^t - \gamma_t\nabla_i^t\right)$, $\gamma_t = \frac{4}{\mu(t+\beta)}$ *and* $\beta = 72L^2/\mu^2$ *(see Step 1 of Algorithm 2).*

*Proof of Lemma 9.* We start with (Allen-Zhu, 2018b, Lemma 3.2), which we state below.

**Lemma 10.** *Allen-Zhu (2018b) If* $w_{t+1} = \arg\min_{y\in\mathbb{R}^d}\{\frac{1}{2\eta}\|y - w_t\|_2^2 + \psi(y) + \langle\tilde{\nabla}_t, y\rangle\}$ *for some random vector* $\tilde{\nabla}_t \in \mathbb{R}^d$ *satisfying* $\mathbb{E}[\tilde{\nabla}_t] = \nabla f(w_t)$, *then for every* $u \in \mathbb{R}^d$, *we have*

$$
\mathbb{E}[F(w_{t+1}) - F(u)] \leq \mathbb{E}\left[\frac{\eta}{2(1 - \eta L)}\left\|\tilde{\nabla}_t - \nabla f(w_t)\right\|_2^2 + \frac{(1 - \mu_f\eta)\|u - w_t\|_2^2 - (1 + \mu_\psi\eta)\|u - w_{t+1}\|_2^2}{2\eta}\right]
$$

*where* $\psi$ *is a proximal term,* $\mu_\psi$ *is its strong convexity and* $\mu_f$ *is the strong convexity of the optimization function* $f$ *and* $\eta$ *is the step of the algorithm and the function* $F$ *is defined as:*

$$
F(x) = f(x) + \psi(x)
$$

For our setting the proximal term is:

$$
\psi(x) = \begin{cases} 0, & x \in \mathcal{D} \\ \infty, & x \notin \mathcal{D} \end{cases}
$$

This means that $\mu_\psi = 0$. The step $\eta$ for our analysis is $\gamma_t$ and $f \equiv g_i$.

By taking $u = x_i^\star = \arg\min_{x\in\mathcal{D}} g_i(x)$ and since due to projection on $\mathcal{D}$

$$
F(x_i^{t+1}) = g_i(x_i^{t+1}) + \psi(x_i^{t+1}) = g_i(x_i^{t+1})
$$

the inequality can be restated as:

$$\mathbb{E}[g_i(x_i^{t+1}) - g_i(x_i^\star)] \leq \mathbb{E}\left[\frac{\gamma_t}{2(1 - \gamma_t L)}\left\|\nabla_i^t - \nabla g_i(x_i^t)\right\|_2^2 + \frac{1 - \mu\gamma_t}{2\gamma_t}\left\|x_i^\star - x_i^t\right\|_2^2 - \frac{1}{2\gamma_t}\left\|x_i^\star - x_i^{t+1}\right\|_2^2\right]$$

Notice that by the selection of $\gamma_t = 4/(\mu(t + \beta))$ in Step 6 of Algorithm 2 we can do the following simplification

$$\frac{1}{(1 - \gamma_t L)} = \frac{1}{(1 - \frac{4L}{\mu(t+\beta)})} \leq \frac{1}{(1 - \frac{4L}{\mu\beta})} = \frac{1}{(1 - \frac{4\mu}{72L})} \leq \frac{1}{(1 - \frac{4}{72})} = \frac{18}{17} \leq \frac{9}{8}$$

which gives the theorem statement. The last part of the proof required, is to show that the following two update rules are equivalent.

$$x_i^{t+1} = \underset{y \in \mathbb{R}^d}{\arg\min}\{\frac{1}{2\gamma_t}\left\|y - x_i^t\right\|_2^2 + \psi(y) + \langle\nabla_i^t, y\rangle\}$$
$$x_i^{t+1} = \Pi_{\mathcal{D}}\left(x_i^t - \gamma_t\nabla_i^t\right)$$

which is a well known fact in the literature and this concludes the proof. □

## F   OMITTED PROOFS OF SECTION 4

### F.1   PROOF OF LEMMA 1

In this section we prove Lemma 1, for the sake of exposition we restate it up next.

**Lemma 1.** *Over a sequence of $n$ stages, Algorithm 1 reaches Step 5 and 12, $\lceil \log n/\alpha \rceil$ times.*

*Proof of Lemma 1.* Step 5 and 12 are only executed when the following inequality is satisfied:

$$i - \text{prev} \geq \alpha \cdot i \Rightarrow i \geq \frac{1}{1 - \alpha} \cdot \text{prev} \tag{11}$$

Once Algorithm 1 reaches Step 5 and 12 it necessarily, reaches Step 13 where prev is updated to $i$. Let $z_0 = 1, z_1, \ldots, z_k, \ldots$ the sequence of stages where $z_k$ denotes the stage at which Algorithm 1 reached Step 5 and 12 for the $k$-th time. By Equation 11 we get that $z_{k+1} \geq \frac{1}{1-\alpha} \cdot z_k$ implying that

$$z_k \geq \left(\frac{1}{1 - \alpha}\right)^k$$

Since $z_k \leq n$ we get that $k \leq \frac{\log n}{\log\left(\frac{1}{1-\alpha}\right)}$. Notice that $\log\left(\frac{1}{1-\alpha}\right) = -\log(1 - \alpha) \geq 1 - (1 - \alpha) = \alpha$ and thus $k \leq \frac{\log n}{\alpha}$. □

Using Lemma 1, we can now also show Corollary 1.

**Corollary 1.** *Over a sequence of $n$ stages, Algorithm 1 requires $3\sum_{i=1}^n T_i + 2n\lceil \log n/\alpha \rceil$ FOs.*

*Proof of Corollary 1.* At each iteration of Algorithm 2 requires 3 FOs (Step 5) and thus Algorithm 2 requires overall $3T_i$ FOs during stage $i \in [n]$. At Step 5 and 12 Algorithm 1 requires at most $n$ FOs and thus by Lemma 1 it overall requires $2n\lceil \log n/\alpha \rceil$ FOs. □

### F.2   PROOF OF THEOREM 4

In this section we provide the proof for Theorem 4, for the sake of exposition we restate it up next.

**Theorem 4.** *Let a convex and compact set $\mathcal{D}$ and a sequence $\mu$-strongly convex functions $f_1, \ldots, f_n$ with $f_i : \mathcal{D} \mapsto \mathbb{R}$. Then Algorithm 1, with $T_i = 720GL^2/(\mu^{5/2}i\sqrt{\epsilon}) + 9L^{2/3}G^{2/3}/(\epsilon^{1/3}\mu) + 864L^2/\mu^2$ and $\alpha = \mu\epsilon^{1/3}/(20G^{2/3}L^{2/3})$, guarantees*

$$\mathbb{E}[g_i(\hat{x}_i)] - g_i(x_i^\star) \leq \epsilon \quad \text{for each stage } i \in [n]$$

*where $\hat{x}_i \in \mathcal{D}$ is the output of Algorithm 2 at Step 9 of Algorithm 1.*

*Proof of Theorem 4.* At stage $i := 1$, Algorithm 1 performs gradient descent using $f_1$ in order to produce $\hat{x}_1 \in \mathcal{D}$. The latter requires $\mathcal{O}\left(L \log(1/\epsilon)/\mu\right)$ FOs. Let us inductively assume that,

$$\mathbb{E}[g_j(\hat{x}_j)] - g_j(x_j^\star) \leq \epsilon \quad \text{for all} \quad j \in [i-1].$$

Using the latter we will establish that $\mathbb{E}[g_i(\hat{x}_i)] - g_i(x_i^\star) \leq \epsilon$. In order to do the latter we first use Lemma 8. The proof of which can be found in Appendix E and its proof is based on the fact that the estimator $\nabla_i^t$ is unbiased and admits bounded variance (see Lemma 2 and 3).

**Lemma 8** (Convergence Bound). *If $\mathbb{E}[g_j(\hat{x}_j)] - g_j(x_j^\star) \leq \epsilon$ for all stages $j \in [i-1]$ then*

$$\mathbb{E}\left[g_i\left(\hat{x}_i\right) - g_i(x_i^\star)\right] \leq \frac{1}{\mathcal{Z}}\left(\frac{\mu}{8}(\beta-1)(\beta-2)\mathbb{E}\left[\left\|x_i^0 - x_i^\star\right\|_2^2\right] + \kappa^2 G^2 \alpha^2 \frac{1}{\mu}T_i + 2\kappa^2 \epsilon T_i\right) \quad (7)$$

*where $\mathcal{Z} = \frac{T_i(T_i-1)}{2} + (T_i+1)(\beta-1)$, $\kappa = 6L/\mu$ and $\beta = 72L^2/\mu^2$.*

Since the conditions of Lemma 8 are ensured by the induction hypothesis, we will appropriately select $T_i$ such that the right-hand side of Equation 7 is upper bound by $\epsilon > 0$.

At first, we upper bound the term $\mathbb{E}\left[\left\|x_i^0 - x_i^\star\right\|_2^2\right]$ appearing in the first term of the RHS of Equation 7. Recall that in Step 2 of Algorithm 2, we set $x_i^0 \leftarrow \hat{x}_{i-1} \in \mathcal{D}$. As a result,

$$\left\|x_i^0 - x_i^\star\right\|_2^2 = \left\|\hat{x}_{i-1} - x_i^\star\right\|_2^2$$

In order to upper bound the term $\left\|\hat{x}_{i-1} - x_i^\star\right\|_2^2$ we use Lemma 6, the proof of which can be found in Appendix B.

**Lemma 6.** *Let $\hat{x}_j \in \mathcal{D}$ the output of Algorithm 1 for stage $j \in [n]$. Then for all $i \in \{j+1, \ldots, n\}$,*

$$\left\|\hat{x}_j - x_i^\star\right\|_2^2 \leq \frac{8}{\mu^2}\left(\frac{G(i-j)}{i+j}\right)^2 + 2\left\|\hat{x}_j - x_j^\star\right\|_2^2$$

Applying Lemma 6 for $j := i-1$ we get that

$$\mathbb{E}\left[\left\|x_i^0 - x_i^\star\right\|_2^2\right] = \mathbb{E}\left[\left\|\hat{x}_{i-1} - x_i^\star\right\|_2^2\right] \leq \frac{8}{\mu^2}\left(\frac{G}{2i-1}\right)^2 + 2\mathbb{E}\left[\left\|\hat{x}_{i-1} - x_{i-1}^\star\right\|_2^2\right] \quad (12)$$

By the inductive hypothesis we know that $\mathbb{E}\left[g_{i-1}(\hat{x}_{i-1})\right] - g_{i-1}(x_{i-1}^\star) \leq \epsilon$ and thus by the strong convexity $g_i(\cdot)$, we get that

$$\mathbb{E}\left[\left\|\hat{x}_{i-1} - x_{i-1}^\star\right\|_2^2\right] \leq \frac{2}{\mu}\epsilon$$

which yields Equation 13:

$$\mathbb{E}\left[\left\|x_i^0 - x_i^\star\right\|_2^2\right] \leq \frac{8}{\mu^2}\left(\frac{G}{2i-1}\right)^2 + \frac{4}{\mu}\epsilon \quad (13)$$

The value of $\mathcal{Z}$ can be lower bounded as follows:

$$\mathcal{Z} := T_i(T_i-1)/2 + (T_i+1)(\beta-1) \geq T_i(T_i-1)/2 \geq T_i^2/4$$

So by combining Equation 7 with the previous two inequalities we get:

$$
\begin{aligned}
\mathbb{E}[g_i(\hat{x}_i)] - g_i(x_i^\star) &\leq \frac{4}{T_i^2}\left(\frac{\mu}{8}(\beta-1)(\beta-2)\left(\frac{8}{\mu^2}\left(\frac{G}{2i-1}\right)^2 + \frac{4}{\mu}\epsilon\right) + \kappa^2 G^2 \alpha^2 \frac{1}{\mu}T_i + 2\kappa^2 \epsilon T_i\right) \\
&= \left(\frac{4G^2}{\mu T_i^2(2i-1)^2} + \frac{2\epsilon}{T_i^2}\right)(\beta-1)(\beta-2) + 4\kappa^2 G^2 \frac{\alpha^2}{T_i \mu} + 8\kappa^2 \epsilon \frac{1}{T_i}
\end{aligned}
$$

Now the upper bound on $\mathbb{E}[g_i(\hat{x}_i)] - g_i(x_i^\star)$ admits four terms all of which depend on $T_i$. Thus we can select $T_i$ so that each of the terms is upper bounded by $\epsilon/4$. Namely,

$$\begin{cases} \frac{4G^2}{\mu T_i^2 (2i-1)^2}(\beta-1)(\beta-2) & \le \epsilon/4 \\ \frac{2\epsilon}{T_i^2}(\beta-1)(\beta-2) & \le \epsilon/4 \\ 4\kappa^2 G^2 \frac{\alpha^2}{i^2 T_i \mu} & \le \epsilon/4 \\ 8\kappa^2 \epsilon \frac{1}{T_i} & \le \epsilon/4 \end{cases}$$

Since $\kappa = 6\frac{L}{\mu}$, we get that we can set $T_i$ as follows,

$$\begin{aligned} T_i &= \max\{192\frac{GL^2}{\mu^{5/2} i \sqrt{\epsilon}}, 204\frac{L^2}{\mu^2}, 576\frac{L^2 G^2 \alpha^2}{\mu^3 i^2 T_i}, 1152\frac{L^2}{\mu^2}\} \\ &= \max\{192\frac{GL^2}{\mu^{5/2} i \sqrt{\epsilon}}, 576\frac{L^2 G^2 \alpha^2}{\mu^3 i^2 T_i}, 1152\frac{L^2}{\mu^2}\} \\ &\le 192\frac{GL^2}{\mu^{5/2} i \sqrt{\epsilon}} + 576\frac{L^2 G^2 \alpha^2}{\mu^3 i^2 T_i} + 1152\frac{L^2}{\mu^2} \end{aligned}$$

The proof is completed by selecting $\alpha = \mu \epsilon^{1/3}/(9 G^{2/3} L^{2/3})$. $\qquad\square$

## G   PROOF OF THEOREM 1

In this section we provide the proof for Theorem 1.

**Theorem 1.** *There exists a first-order method,* CSVRG *(Algorithm 1), for continual finite-sum minimization (2) with* $\mathcal{O}\left(\frac{L^{2/3} G^{2/3}}{\mu} \cdot \frac{n \log n}{\epsilon^{1/3}} + \frac{L^2 G}{\mu^{5/2}} \cdot \frac{\log n}{\sqrt{\epsilon}}\right)$ *FO complexity.*

*Proof of Theorem 1.* The proof of Theorem 1 follows by combining Theorem 4 and Corollary 1. Their proofs are respectively in Appendix F.2 and F.1.

**Theorem 4.** *Let a convex and compact set $\mathcal{D}$ and a sequence $\mu$-strongly convex functions $f_1, \ldots, f_n$ with $f_i : \mathcal{D} \mapsto \mathbb{R}$. Then Algorithm 1, with $T_i = 720 GL^2/(\mu^{5/2} i \sqrt{\epsilon}) + 9L^{2/3}G^{2/3}/(\epsilon^{1/3}\mu) + 864 L^2/\mu^2$ and $\alpha = \mu\epsilon^{1/3}/(20 G^{2/3} L^{2/3})$, guarantees*

$$\mathbb{E}\left[g_i(\hat{x}_i)\right] - g_i(x_i^\star) \le \epsilon \quad \text{for each stage } i \in [n]$$

*where $\hat{x}_i \in \mathcal{D}$ is the output of Algorithm 2 at Step 9 of Algorithm 1.*

**Corollary 1.** *Over a sequence of $n$ stages, Algorithm 1 requires $3\sum_{i=1}^{n} T_i + 2n\lceil \log n/\alpha \rceil$ FOs.*

Using the selection of $T_i$ provided in Theorem 4 to Corollary 1 we get that

$$\begin{aligned} \sum_{i=1}^{n} T_i &\le \frac{720 GL^2}{\mu^{5/2} \sqrt{\epsilon}} \sum_{i=1}^{n} \frac{1}{i} + n\left(\frac{9L^{2/3} G^{2/3}}{\epsilon^{1/3}\mu} + 864\frac{L^2}{\mu^2}\right) \\ &\le \frac{720 GL^2}{\mu^{5/2} \sqrt{\epsilon}} \log n + n\left(\frac{9L^{2/3} G^{2/3}}{\epsilon^{1/3}\mu} + 864\frac{L^2}{\mu^2}\right) \end{aligned}$$

At the same time, using the selection of $\alpha$ provided in Theorem 4 we get

$$2n\log n/\alpha := 40\frac{L^{2/3} G^{2/3}}{\mu}\frac{n \log n}{\epsilon^{1/3}}$$

$\qquad\square$

## H   PROOF OF THEOREM 3

In this section we provide the proof of Theorem 3. We first present an intermediate theorem that is the main technical contribution of the section.

**Theorem 5.** *Let a natural first-order method $\mathcal{A}$ (even randomized) that given a sequence of $n$ strongly convex functions $f_1, \ldots, f_n$ outputs a sequence $\hat{x}_1, \ldots, \hat{x}_n \in \mathcal{D}$ by performing overall $o(n^2)$ FOs. Then there exists a sequence of strongly convex functions $f_1, \ldots, f_n$ with $f_i : [-1, 1]^d \mapsto \mathbb{R}$ and $\mu, G, L = \mathcal{O}(1)$ such that*

$$\mathbb{E}\left[g_i(\hat{x}_i)\right] - g_i(x_i^\star) \geq \Omega\left(1/n^4\right) \quad \text{for some stage } i \in [n]$$

The proof of Theorem 5 lies in Section H.1. To this end we use Theorem 5 to establish Theorem 3.

**Theorem 3.** *For any $\alpha > 0$, there is no natural first-order method for Problem (2) with $\mathcal{O}\left(n^{2-\alpha} \log(1/\epsilon)\right)$ FO complexity. Moreover for any $\alpha < 1/4$, there is no natural first-order method for continual finite-sum minimization (2) with $\mathcal{O}\left(n/\epsilon^\alpha\right)$ FO complexity.*

*Proof of Theorem 3.* Let us assume that there exists a natural first order method $\mathcal{A}$ with overall complexity $\mathcal{O}\left(n^{2-\alpha} \log(1/\epsilon)\right)$ for some $\alpha > 0$. By setting $\epsilon = \mathcal{O}(1/n^5)$, we get that there exists a natural first-order method that for sequence $f_1, \ldots, f_n$ guarantees

$$\mathbb{E}\left[g_i(\hat{x}_i)\right] - g_i(x_i^\star) \leq \mathcal{O}\left(1/n^5\right) \quad \text{for each stage } i \in [n]$$

with overall FO complexity $\mathcal{O}(n^{2-\alpha} \log n)$. However the latter contradicts with Theorem 5.

Respectively let us assume that there exists a natural first order method $\mathcal{A}$ with overall complexity $\mathcal{O}\left(n/\epsilon^\alpha\right)$ for some $\alpha < 1/4$. By setting $\epsilon = \mathcal{O}(1/n^4)$, we get that there exists a natural first-order method that for sequence $f_1, \ldots, f_n$ guarantees

$$\mathbb{E}\left[g_i(\hat{x}_i)\right] - g_i(x_i^\star) \leq \mathcal{O}\left(1/n^4\right) \quad \text{for each stage } i \in [n]$$

with overall FO complexity $\mathcal{O}(n^{1+\alpha/4})$. However the latter contradicts with Theorem 5. □

## H.1 Proof of Theorem 5

*Proof of Theorem 5.* To simplify notation, we denote with $[x]_\ell$ the coordinate $\ell$ of vector $x \in \mathbb{R}^d$. In our lower bound construction we consider $d := 2$.

Since $\mathcal{A}$ performs $o(n^2)$ FOs then there exists a stage $i > 1$ such that $\sum_{t \in T_i} |Q_i^t| \leq i/2$ (otherwise the overall number of FOs, $\sum_{i \in [n]} \sum_{t \in T_i} |Q_i^t| \geq n^2/4$). Using this specific index $i \in [n]$ we construct the following (random) sequence of functions $f_1, \ldots, f_n$ where each $f_i : [-1, 1]^2 \mapsto \mathbb{R}$:

$$f_\ell(x) = \begin{cases} ([x]_1)^2 + ([x]_2)^2 & \ell \neq i, k \\ ([x]_1)^2 + ([x]_2)^2 + ([x]_1 - [x]_2)^2 & \ell = k \\ ([x]_1 - 1)^2 + ([x]_1)^2 + ([x]_2)^2 & \ell = i \end{cases}$$

where $k \sim \text{Unif}(1, \ldots, i-1)$.

Before proceeding we remark that each function $f_\ell : [-1, 1]^2 \mapsto \mathbb{R}$ is $G$-Lipschitz, $L$-smooth and $\mu$-strongly convex with $G, L, \mu = \mathcal{O}(1)$.

**Corollary 2.** *Each $f_\ell : [-1, 1]^2 \mapsto \mathbb{R}$ is 2-strongly convex, 6-smooth and $6\sqrt{2}$-Lipschitz.*

Let the initial point of $\mathcal{A}$ be $\hat{x}_0 = (0, 0) \in [-1, 1]^2$.

Notice that $\nabla f_\ell(0, 0) = (0, 0)$ for each $\ell \neq i$. Since $\mathcal{A}$ is a natural first-order method then Definition 3 implies that $\mathcal{A}$ always remains that $(0, 0)$ at all stages $j \leq i - 1$. The latter is formally stated and established in Lemma 11.

**Lemma 11.** *For any stage $j \in [i-1]$,*

- $[\hat{x}_j]_1 = [\hat{x}_j]_2 = 0$

- $[x_j^t]_1 = [x_j^t]_2 = 0$ *for all rounds $t \in [T_j]$.*

The proof of Lemma 11 lies in Section H.2. Its proof is based on the simple observation that if $\mathcal{A}$ has stayed at $(0,0)$ at all stages $\ell \leq i - 2$. Then at stage $\ell + 1 \leq i - 1$, Item 1 of Definition 3 implies that $\mathcal{A}$ always queries $\nabla f_j(0,0)$ for some $j \leq i - 1$ meaning that $\nabla f_j(0,0) = (0,0)$. Then Item 2 and 3 of Definition 3 directly imply that $x_\ell^t = (0,0)$ and $\hat{x}_\ell = (0,0)$.

**Corollary 3.** *With probability greater than $1/2$, the function $f_k$ is never queried during state $i$. In other words, $q_{index} \neq k$ for all $q \in \cup_{t \in T_i} Q_i^t$.*

*Proof.* By the selection of stage $i \in [n]$, we know that $\sum_{t \in T_i} |Q_i^t| \leq i/2$. Since $k$ was sampled uniformly at random in $\{1, \ldots, i - 1\}$, $\Pr\left[q_{index} \neq k \text{ for all } q \in \cup_{t \in T_i} Q_i^t\right] \geq 1/2$. $\qquad\square$

Notice that for all $\ell \neq i$, $\nabla f_\ell(\cdot, 0) = (\cdot, 0)$. The main idea of the proof is that in case $k$ is never queried during stage $i$, ($q_{index} \neq k$ for all $q \in \cup_{t \in T_i} Q_i^t$) then all FOs during stage admit the form $\nabla f_{index}(q_{value}) = \nabla f_{index}(\cdot, 0) = (\cdot, 0)$. The latter implies that if $q_{index} \neq k$ for all $q \in \cup_{t \in T_i} Q_i^t$ then $[\hat{x}_i]_2 = 0$.

**Lemma 12.** *In case $q_{index} \neq k$ for all $q \in \cup_{t \in T_i} Q_i^t$ then $[\hat{x}_i]_2 = 0$.*

The proof of Lemma 12 is based on the intuition that we presented above. Its proof is presented in Appendix H.2.

To this end we are ready to complete the proof of Theorem 5. Combining Lemma 12 with Corollary 3 we get that

$$\Pr\left[[\hat{x}_i]_2 = 0\right] \geq 1/2. \tag{14}$$

Up next we lower bound the error $g_i(\hat{x}_i) - g_i(x_i^\star)$ in case $[\hat{x}_i]_2 = 0$.

By the definition of the sequence $f_1, \ldots, f_k, \ldots, f_i$, we get that

$$g_i(x) \quad := \quad ([x]_1)^2 + \frac{1}{i}([x]_1 - 1)^2 + \frac{1}{i}([x]_1 - [x]_2)^2 + ([x]_2)^2$$

In case $[\hat{x}_i]_2 = 0$ we get that

$$
\begin{aligned}
g_i(\hat{x}_i) &= \left(1 + \frac{1}{i}\right)([\hat{x}_i]_1)^2 + \frac{1}{i}([\hat{x}_i]_1 - 1)^2 \\
&\geq \min_{w \in [-1,1]}\left[\left(1 + \frac{1}{i}\right)w^2 + \frac{1}{i}(w-1)^2\right] \quad \text{notice that } w^\star = \frac{1}{i+2} \\
&= \frac{i+1}{i(i+2)^2} + \frac{(i+1)^2}{i(i+2)^2} = \frac{i+1}{i(i+2)}
\end{aligned}
$$

At the same time, the minimum $g_i(x_i^\star)$ equals,

$$
\begin{aligned}
g_i(x_i^\star) &= ([x_i^\star]_1)^2 + \frac{1}{i}([x_i^\star]_1 - 1)^2 + \frac{1}{i}([x_i^\star]_1 - [x_i^\star]_2)^2 + ([x_i^\star]_2)^2 \\
&= \min_{w,z \in [-1,1]}\left[w^2 + \frac{1}{i}(w-1)^2 + \frac{1}{i}(w-z)^2 + z^2\right] \quad \text{notice that } (w^\star, z^\star) = \left(\frac{i+1}{i^2 + 3i + 1}, \frac{1}{i^2 + 3i + 1}\right) \\
&= \left(\frac{i+1}{i^2 + 3i + 1}\right)^2 + \frac{1}{i}\left(\frac{i+1}{i^2 + 3i + 1} - 1\right)^2 + \frac{1}{i}\left(\frac{i+1}{i^2 + 3i + 1} - \frac{1}{i^2 + 3i + 1}\right)^2 + \left(\frac{1}{i^2 + 3i + 1}\right)^2
\end{aligned}
$$

By taking the difference $g_i(\hat{x}_i) - g_i(x_i^\star)$ we get that

$$
\begin{aligned}
g_i(\hat{x}_i) - g_i(x_i^\star) &\geq \min_{w \in [-1,1]}\left[w^2 + \frac{1}{i}(w-1)^2\right] - \min_{w,z \in [-1,1]^2}\left[\left(1 - \frac{1}{i}\right)w^2 + \frac{1}{i}(w-1)^2 + \frac{1}{i}(w-z)^2\right] \\
&= \frac{i+1}{i(i+2)} - \left(\frac{i+1}{i^2 + 3i + 1}\right)^2 - \frac{1}{i}\left(\frac{i+1}{i^2 + 3i + 1} - 1\right)^2 - \frac{1}{i}\left(\frac{i+1}{i^2 + 3i + 1} - \frac{1}{i^2 + 3i + 1}\right)^2 \\
&\quad - \left(\frac{1}{i^2 + 3i + 1}\right)^2 \\
&= \frac{1}{i^4 + 5i^3 + 7i^2 + 2i}
\end{aligned}
$$

We finally get that

$$
\begin{aligned}
\mathbb{E}\left[g_i(\hat{x}_i) - g_i(x_i^\star)\right] &= \Pr\left[[\hat{x}_i]_2 = 0\right] \cdot \mathbb{E}\left[g_i(\hat{x}_i) - g_i(x_i^\star) \mid [\hat{x}_i]_2 = 0\right] \\
&+ \Pr\left[[\hat{x}_i]_2 \neq 0\right] \cdot \mathbb{E}\left[g_i(\hat{x}_i) - g_i(x_i^\star) \mid [\hat{x}_i]_2 \neq 0\right] \\
&\geq \frac{1}{2} \cdot \frac{1}{i^4 + 5i^3 + 7i^2 + 2i} \\
&\geq \Omega\left(1/n^4\right)
\end{aligned}
$$

$\square$

## H.2 OMITTED PROOFS

**Corollary 2.** *Each $f_\ell : [-1,1]^2 \mapsto \mathbb{R}$ is 2-strongly convex, 6-smooth and $6\sqrt{2}$-Lipschitz.*

*Proof.* Notice that Hessians

- $\nabla^2 f_\ell(x) = \begin{pmatrix} 2 & 0 \\ 0 & 2 \end{pmatrix}$ for $\ell \neq i, k$. Meaning that $f_\ell$ is 2-strongly convex and 2-smooth.

- $\nabla^2 f_k(x) = \begin{pmatrix} 4 & -2 \\ -2 & 4 \end{pmatrix}$ for $\ell \neq i, k$. Meaning that $f_\ell$ is 2-strongly convex and 6-smooth.

- $\nabla^2 f_i(x) = \begin{pmatrix} 4 & 0 \\ 0 & 2 \end{pmatrix}$ for $\ell \neq i, k$. Meaning that $f_\ell$ is 2-strongly convex and 4-smooth.

Respectively notice that the gradients

- $\nabla f_\ell(x) = 2([x]_1, [x]_2)$ and thus $\max_{[x]_1, [x]_2 \in [-1,1]} \|\nabla f_\ell(x)\|_2 \leq 2\sqrt{2}$. As a result, $f_\ell(x)$ is $2\sqrt{2}$-Lipschtiz in $[-1,1]^2$.

- $\nabla f_k(x) = 2(2[x]_1 - [x]_2, 2[x]_2 - [x]_1)$ and thus $\max_{[x]_1, [x]_2 \in [-1,1]} \|\nabla f_k(x)\|_2 \leq 6\sqrt{2}$. As a result, $f_k(x)$ is $6\sqrt{2}$-Lipschtiz in $[-1,1]^2$.

- $\nabla f_i(x) = 2(2[x]_1 - 1, 2[x]_2)$ and thus $\max_{[x]_1, [x]_2 \in [-1,1]} \|\nabla f_i(x)\|_2 \leq 4$. As a result, $f_k(x)$ is 4-Lipschtiz in $[-1,1]^2$.

$\square$

**Lemma 11.** *For any stage $j \in [i-1]$,*

- $[\hat{x}_j]_1 = [\hat{x}_j]_2 = 0$
- $[x_j^t]_1 = [x_j^t]_2 = 0$ *for all rounds $t \in [T_j]$.*

*Proof of Lemma 11.* By the construction of the sequence $f_1, \ldots, f_n$, we get that for all $\ell \leq i-1$,

- $\nabla f_\ell(x) = 2([x]_1, [x]_2)$ for $\ell \neq k, i$
- $\nabla f_k(x) = 2(2[x]_1 - [x]_2, 2[x]_2 - [x]_1)$

Thus in case $x = (0,0)$ then $\nabla f_\ell(x) = \nabla f_k(x) = (0,0)$.

Up next we inductively establish Lemma 11. For the induction base we first establish that at stage $\ell := 1$,

**Corollary 4.** *For stage $\ell = 1$,*

- $x_1^t = (0,0)$ *for all rounds $t \in [T_1]$.*

- $\hat{x}_1 = (0,0)$

*Proof.* At round $t := 1$ of stage $\ell := 1$, Item 1 of Definition 3 ensures that for all FOs $q \in Q_1^1$

$$q_{\text{index}} = 1 \text{ and } q_{\text{value}} = (0,0)$$

Since $\nabla f_1(x) = 2x$, the latter implies that $\nabla f_{q_{\text{index}}}(q_{\text{value}}) = (0,0)$ for all $q \in Q_1^1$. Thus Item 2 of Definition 3 ensures that $x_1^1 = (0,0)$.

We prove Corollary 4 through induction.

- Induction Base: $x_1^1 = (0,0)$

- Induction Hypothesis: $x_1^\tau = (0,0)$ for all $\tau \in \{1, \ldots, t-1\}$.

- Induction Step: $x_1^t = (0,0)$

Item 1 of Definition 3 ensures that for all $q \in Q_1^t$, $q_{\text{value}} \in (\cup_{\tau \le t-1} x_1^\tau) \cup \hat{x}_0$ and thus by the inductive hypothesis $q_{\text{value}} = (0,0)$. Thus

$$\nabla f_{q_{\text{index}}}(q_{\text{value}}) = (0,0) \text{ for all } q = (q_{\text{index}}, q_{\text{value}}) \in \cup_{\tau \le t} Q_1^\tau$$

Then Item 2 of Definition 3 implies that $x_1^t = (0,0)$.

To this end we have established that $x_1^t = (0,0)$ for all $t \in [T_1]$. Then Item 3 of Definition 3 implies that $\hat{x}_1 = (0,0)$. □

We complete the proof of Lemma 11 with a similar induction.

- Induction Base: $x_1^t = (0,0)$ for all $t \in [T_1]$ and $\hat{x}_1 = (0,0)$.

- Induction Hypothesis: $x_\ell^t = (0,0)$ for all $t \in [T_\ell]$ and $\hat{x}_\ell = (0,0)$, for all stages $\ell \le i-2$.

- Induction Step: $x_{\ell+1}^t = (0,0)$ for all $t \in [T_{\ell+1}]$ and $\hat{x}_{\ell+1} = (0,0)$

Let us start with round $t := 1$ of stage $\ell + 1$. Item 1 of Definition 3 together with the inductive hypothesis ensure that for any FO $q \in Q_{\ell+1}^1$, $q_{\text{value}} = (0,0)$. Since $q_{\text{index}} \le \ell + 1 \le i - 1$ we get that

$$\nabla f_{q_{\text{index}}}(q_{\text{value}}) = (0,0) \text{ for all queries } q \in Q_{\ell+1}^1.$$

The latter together with Item 2 of Definition 3 imply

$$x_{\ell+1}^1 = (0,0).$$

Let us inductively assume that $x_{\ell+1}^t = (0,0)$. Then we the exact same argument as before we get that

$$\nabla f_{q_{\text{index}}}(q_{\text{value}}) = (0,0) \text{ for all queries } q \in Q_{\ell+1}^{t+1}.$$

Then again Item 2 of Definition 3 implies that

$$x_{\ell+1}^{t+1} = (0,0).$$

To this end we have established that $x_{\ell+1}^t = (0,0)$ for all $t \in [T_{\ell+1}]$. Thus Item 3 of Definition 3 implies that $\hat{x}_{\ell+1} = (0,0)$. The latter completes the induction step and the proof of Lemma 11. □

**Lemma 12.** *In case $q_{\text{index}} \ne k$ for all $q \in \cup_{t \in T_i} Q_i^t$ then $[\hat{x}_i]_2 = 0$.*

*Proof of Lemma 12.* First notice that

- $\nabla f_\ell(x) = 2([x]_1, [x]_2)$ for $\ell \ne k, i$

- $\nabla f_i(x) = 2(2[x]_1 - 1, [x]_2)$

Thus for any $x = ([x]_1, 0)$, $\nabla f_\ell(x) = 2([x]_1, 0)$ and $\nabla f_i(x) = 2(2[x]_1 - 1, 0)$. Using the latter observation and the Lemma 11 we inductively establish that in case $q_{\text{index}} \neq k$ for all $q \in \cup_{t \in T_i} Q_i^t$ then

$$x_i^t = \left([x_i^t]_1, 0\right).$$

We start by establishing the latter for $t = 1$.

**Corollary 5.** *In case $q_{\text{index}} \neq k$ for all $q \in \cup_{t \in T_i} Q_i^t$ then $x_i^t = ([x_i^t]_1, 0)$.*

*Proof.* Lemma 11 ensures that for all stages $\ell \leq i - 1$

- $x_\ell^t = (0, 0)$ for all $t \in [T_\ell]$.

- $\hat{x}_\ell = (0, 0)$

The latter together with Item 1 of Definition 3 imply that $q_{\text{value}} = (0, 0)$ for all $q \in Q_i^1$. Since $q_{\text{index}} \neq k$ for all $q \in Q_i^1$ we get that

$$\nabla f_{q_{\text{index}}}(q_{\text{value}}) = (\cdot, 0) \text{ for all } q \in Q_i^1$$

Then Item 2 of Definition 3 implies that $x_i^1 = (\cdot, 0)$. $\qquad\square$

We complete the proof of Lemma 12 through an induction.

- Induction Base: $x_i^1 = (\cdot, 0)$

- Induction Hypothesis: $x_i^\tau = (\cdot, 0)$ for all $\tau \leq t - 1$

- Induction Step: $x_i^{t+1} = (\cdot, 0)$

The induction hypothesis together with Lemma 11 and Item 1 of Definition 3 imply that for each FO $q \in Q_i^t$, $q_{\text{value}} = (\cdot, 0)$. Since $k \neq q_{\text{value}}$ for all $q \in Q_i^t$, we get that

$$\nabla f_{q_{\text{index}}}(q_{\text{value}}) = (\cdot, 0) \text{ for all } q \in Q_i^t$$

Then Item 2 of Definition 3 implies that $x_i^t = (\cdot, 0)$.

To this end our induction is completed and we have established that $x_i^t = (\cdot, 0)$ for all $t \in [T_i]$. Then the fact that $\hat{x}_i = (\cdot, 0)$ is directly implied by Item 3 of Definition 3.

$\qquad\square$

# I    SPARSE STOCHASTIC GRADIENT DESCENT

In this section we present the sparse version Stochastic Gradient Descent, called SGD-Sparse (Algorithm 4) that is able to ensure accuracy $\epsilon > 0$ at each stage $i \in [n]$ with $\mathcal{O}\left(\frac{\mathcal{D}G^3 \log n}{\epsilon^2}\right)$ FO complexity.

In Algorithm 3 we first present SGD for specific stage $i \in [n]$.

---

**Algorithm 3** StochasticGradientDescent$(i, x, \gamma, T_i)$

---

1: $x_0^i \leftarrow x \in \mathcal{D}$
2: **for** each round $t := 1, \ldots, T_i$ **do**
3:      Sample $j_t \sim \text{Unif}(\{1, \ldots, i-1\})$
4:      $x_i^t \leftarrow \Pi_{\mathcal{D}}\left(x_i^{t-1} - \gamma \nabla f_{j_t}(x_i^{t-1})/t\right)$
5: **end for**
6: **return** $\hat{x}_i \leftarrow \sum_{t=1}^{T_i} x_i^t / T_i$

---

SGD-sparse is presented in Algorithm 4. Algorithm 4 tracks a sparse sub-sequence of stages $i \in [n]$ (prev $\cdot (1 + \alpha) < i$) at which the returned point $\hat{x}_i$ is produced by running SGD (Algorithm 3) for $T_i$ iterations. In the remaining epochs Algorithm 4 just returns the solution of the last stage at which SGD was used ($\hat{x}_i \leftarrow \hat{x}_{\text{prev}}$, Step 8 of Algorithm 4). Notice that the sparsity of the SGD-call (Step 4) is controlled by the selection of the parameter $\alpha \geq 0$. For example for $\alpha = 0$, Step 4 is reached at each stage $i \in [n]$ while for large values of $\alpha \geq 0$, Step 4 might never be reached.

---

**Algorithm 4** StochasticGradientDescent-sparse

---

1: $\hat{x}_0 \in \mathcal{D}$ and prev $\leftarrow 0$
2: **for** each stage $i := 1, \ldots, n$ **do**
3:      **if** prev $\cdot (1 + \alpha) < i$ **then**
4:          $\hat{x}_i \leftarrow$ StochasticGradientDescent$(i, \hat{x}_{i-1}, \gamma, T_i)$   *# output $\hat{x}_i \in \mathcal{D}$ for stage $i$*
5:          $\hat{x}_{\text{prev}} \leftarrow \hat{x}_i$
6:          prev $\leftarrow i$
7:      **else**
8:          $\hat{x}_i \leftarrow \hat{x}_{\text{prev}}$   *# output $\hat{x}_i \in \mathcal{D}$ for stage $i$*
9:      **end if**
10: **end for**

---

In the rest of the section we establish the formal guarantees of SGD that are formally stated in Algorithm 4. We start by presented the well-known guarantees of SGD.

**Theorem 6** (Hazan (2023)). *Let $\hat{x}_i :=$ StochasticGradientDescent$(i, \hat{x}_{i-1}, \gamma, T_i)$ where $\gamma = 1/\mu$, $T_i = \mathcal{O}\left(\frac{G^2}{\epsilon\mu}\right)$ and $G = \max_{x \in \mathcal{D}} \|\nabla f(x)\|$. Then, $\text{E}\left[g_i(\hat{x}_i) - g_i(x_i^\star)\right] \leq \epsilon$.*

**Theorem 7.** *Let a convex and compact set $\mathcal{D}$ and a sequence of $\mu$-strongly convex functions $f_1, \ldots, f_n$ with $f_i : \mathcal{D} \mapsto \mathbb{R}$. Then Algorithm 4, with $T_i = \mathcal{O}\left(\frac{G^2}{\epsilon\mu}\right)$, $\alpha = \frac{1}{2|\mathcal{D}|G}$ and $\gamma = 1/\mu$, guarantees*

$$\text{E}\left[g_i(\hat{x}_i) - g_i(x_i^\star)\right] \leq \epsilon \text{ for each stage } i \in [n]$$

*The FO complexity of Algorithm 4 is $\mathcal{O}\left(\frac{|\mathcal{D}|G^3 \log n}{\epsilon^2 \mu}\right)$.*

*Proof.* In case $i = \text{prev}$ then by selecting $T = \mathcal{O}\left(\frac{G^2}{\mu\epsilon}\right)$ and $\gamma = 1/\mu$, Theorem 6 implies that $\mathbb{E}\left[g_i(\hat{x}_i)\right] - g_i(x_i^\star) \le \epsilon/2$. Up next we establish the claim for $\text{prev} < i \le (1+\alpha) \cdot \text{prev}$.

$$
\begin{aligned}
\mathbb{E}\left[g_i(\hat{x}_i)\right] - g_i(x_i^\star) &= \mathbb{E}\left[g_i(\hat{x}_{\text{prev}})\right] - g_i(x_i^\star) \quad (\hat{x}_i = \hat{x}_{\text{prev}}, \text{ Step 8 of Algorithm 4}) \\
&= \mathbb{E}\left[\frac{1}{i}\sum_{j=1}^{i} f_j(\hat{x}_{\text{prev}})\right] - \frac{1}{i}\sum_{j=1}^{i} f_j(x_i^\star) \\
&= \frac{\text{prev}}{i} \cdot \mathbb{E}\left[g_{\text{prev}}(\hat{x}_{\text{prev}})\right] + \frac{1}{i}\sum_{j=\text{prev}+1}^{i} \mathbb{E}\left[f_j(\hat{x}_{\text{prev}})\right] - \frac{\text{prev}}{i} \cdot g_{\text{prev}}(x_i^\star) - \frac{1}{i}\sum_{j=\text{prev}+1}^{i} f_j(x_i^\star) \\
&= \frac{\text{prev}}{i} \cdot \left(\mathbb{E}\left[g_{\text{prev}}(\hat{x}_{\text{prev}})\right] - g_{\text{prev}}(x_i^\star)\right) + \frac{1}{i}\sum_{j=\text{prev}+1}^{j}\left(\mathbb{E}\left[f_j(\hat{x}_{\text{prev}})\right] - f_j(x_i^\star)\right) \\
&\le \frac{\text{prev}}{i} \cdot \left(\mathbb{E}\left[g_{\text{prev}}(\hat{x}_{\text{prev}})\right] - g_{\text{prev}}(x_i^\star)\right) + \frac{i - \text{prev}}{i} \cdot |\mathcal{D}|G \\
&= \frac{\text{prev}}{i}\left(\mathbb{E}\left[g_{\text{prev}}(\hat{x}_{\text{prev}})\right] - g_{\text{prev}}(x_{\text{prev}}^\star) + \underbrace{g_{\text{prev}}(x_{\text{prev}}^\star) - g_{\text{prev}}(x_i^\star)}_{\le 0}\right) + |\mathcal{D}|G \cdot \frac{i - \text{prev}}{\text{prev}} \\
&\le \frac{\text{prev}}{i}\left(\mathbb{E}\left[g_{\text{prev}}(\hat{x}_{\text{prev}})\right] - g_{\text{prev}}(x_{\text{prev}}^\star)\right) + |\mathcal{D}|G \cdot \frac{i - \text{prev}}{\text{prev}} \\
&\le \frac{\epsilon}{2} + |\mathcal{D}|G \cdot \frac{i - \text{prev}}{\text{prev}} \\
&\le \frac{\epsilon}{2} + |\mathcal{D}|G\frac{(1+\alpha)\text{prev} - \text{prev}}{\text{prev}} \\
&= \frac{\epsilon}{2} + |\mathcal{D}|G\alpha = \epsilon
\end{aligned}
$$

To this end we have established that $\mathbb{E}\left[g_i(\hat{x}_i)\right] - g_i(x_i^\star) \le \epsilon$ for each stage $i \in [n]$. We conclude the proof by upper bounding the overall FO complexity of Algorithm 4.

An execution of Stochastic Gradient Decent at Step 4 of Algorithm 4 that calculates an $\epsilon/2$ optimal solution requires $\mathcal{O}\left(G^2/(\mu\epsilon)\right)$ FOs. Let us now upper bound the required executions of SGD. Let $k$ denotes number of times Algorithm 4 reached Step 4 over the course of $n$ stages. Due Step 5 of Algorithm 4,

$$
\left(1 + \frac{\epsilon}{2|\mathcal{D}|G}\right)^k \le n \Rightarrow k \le \frac{\ln n}{\ln(1 + \frac{\epsilon}{2|\mathcal{D}|G})} \le \frac{\ln n}{1 - \frac{1}{1+\frac{\epsilon}{2|\mathcal{D}|G}}} \le 4|\mathcal{D}|G\frac{\log n}{\epsilon}
$$

Thus the overall FO complexity of Algorithm 4 is $\mathcal{O}\left(\frac{G^3|\mathcal{D}|\log n}{\mu\epsilon^2}\right)$ $\qquad\square$

# J EXPERIMENTAL DETAILS

In this section we present additional experimental evaluations in LIBSVM datasets.

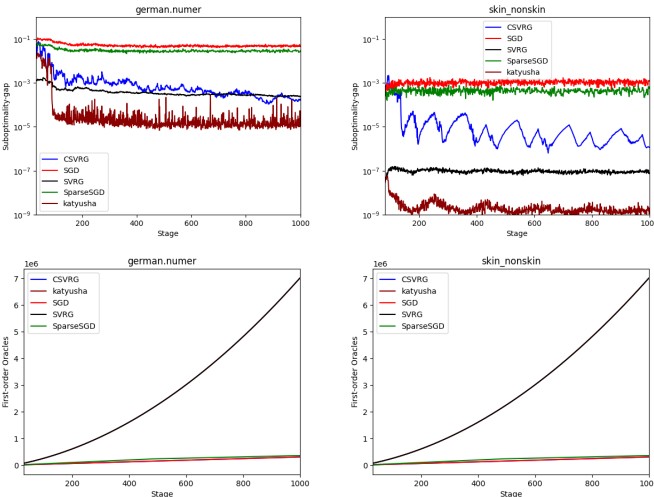

Table 3: Optimality gap as the stages progress on a ridge regression problem (averaged over 10 independent runs). CSVRG performs the exact same number of FOs with SGD and slightly less than SGD-sparse. Katyusha and SVRG perform the exact same number of FOs. CSVRG/SGD/SGD-sparse perform roughly 4% of the FOs of Katyusha/SVRG.

In Table 4 and 5 we present the parameters of the various method used in our experimental evaluations. We remind that $\lambda = 10^{-3}$.

| Method | breast cancer | cod-rna | diabetes |
|---|---|---|---|
| SGD | step size: $(t\lambda)^{-1}$ 
 $T$: 300 | step size: $(t\lambda)^{-1}$ 
 $T$: 300 | step size: $(t\lambda)^{-1}$ 
 $T$: 300 |
| SGD-sparse | step size: $(t\lambda)^{-1}$ 
 $T$: 414 
 $\alpha : 0.002$ | step size: $(t\lambda)^{-1}$ 
 $T$: 480 
 $\alpha : 0.002$ | step size: $(t\lambda)^{-1}$ 
 $T$: 480 
 $\alpha : 0.002$ |
| Katyusha | step size: $1/(3L)$ 
 Outer Iterations: 10 
 Inner Iterations: 100 
 $L : 0.0522$ | step size: $1/(3L)$ 
 Outer Iterations: 10 
 Inner Iterations: 100 
 $L : 0.015$ | step size: $1/(3L)$ 
 Outer Iterations: 10 
 Inner Iterations: 100 
 $L : 0.01262$ |
| SVRG | step size: $1/(3L)$ 
 Outer Iterations: 10 
 Inner Iterations: 100 
 $L : 0.0522$ | step size: $1/(3L)$ 
 Outer Iterations: 10 
 Inner Iterations: 100 
 $L : 0.015$ | step size: $1/(3L)$ 
 Outer Iterations: 10 
 Inner Iterations: 100 
 $L : 0.01262$ |
| CSVRG | step size at $i$: $(it\lambda)^{-1}$ 
 $T_i = 100$ 
 $\alpha = 0.3$ | step size at $i$: $(it\lambda)^{-1}$ 
 $T_i = 100$ 
 $\alpha = 0.3$ | step size at $i$: $(it\lambda)^{-1}$ 
 $T_i = 100$ 
 $\alpha = 0.3$ |

Table 4: Parameters used in our experiments

| Method | german.numer | skin-nonskin |
|---|---|---|
| SGD | step size: $(t\lambda)^{-1}$
$T$: 300 | step size: $(t\lambda)^{-1}$
$T$: 300 |
| SGD-sparse | step size: $(t\lambda)^{-1}$
$T$: 414
$\alpha$ : 0.002 | step size: $(t\lambda)^{-1}$
$T$: 480
$\alpha$ : 0.002 |
| Katyusha | step size: $1/(3L)$
Outer Iterations: 10
Inner Iterations: 100
$L$ : 0.0317 | step size: $1/(3L)$
Outer Iterations: 10
Inner Iterations: 100
$L$ : 0.068 |
| SVRG | step size: $1/(3L)$
Outer Iterations: 10
Inner Iterations: 100
$L$ : 0.0317 | step size: $1/(3L)$
Outer Iterations: 10
Inner Iterations: 100
$L$ : 0.068 |
| CSVRG | step size at $i$: $(it\lambda)^{-1}$
$T_i = 100$
$\alpha = 0.3$ | step size at $i$: $(it\lambda)^{-1}$
$T_i = 100$
$\alpha = 0.3$ |

Table 5: Parameters used in our experiments

## K  EXPERIMENTAL EVALUATIONS WITH NEURAL NETWORKS

In this section we include additional experimental evaluations `CSVRG`,`SGD` and `SVRG` for the continual finite-sum setting in the context of 2-Layer Neural Networks and the MNIST dataset. More precisely, we experiment with Neural Networks with an input layer of size 784 (corresponds to the size of the inputs from the MNIST dataset), a hidden layer with 1000 nodes and an ReLU activation function and an output layer with 10 nodes corresponding to the 10 MNIST classes. As a training loss we use the cross-entropy between the model's output and the one-hot encoding of the classes. We consider two different constructions of the data streams that are respectively presented in Section K.1 and Section K.2. For both settings the parameters used for the various methods are presented in Table K. We also remark that the parameters of `CSVRG`, `SGD` were appropriately selected so as to perform roughly the same number of FOs.

| Method | **MNIST** |
|--------|-----------|
| SGD | step size: 0.001 |
|  | $T$: 1385 |
| SVRG | step size: 0.05 |
|  | Outer Iterations: 7 |
|  | Inner Iterations: 40 |
| CSVRG | step size at $i$: 0.001 |
|  | $T_i = 600$ |
|  | $\alpha = 0.01$ |

Table 6: Parameters used for training of the neural network.

### K.1  DATA STREAMS PRODUCED WITH THE STATIONARY DISTRIBUTION

In this experiment we create a stream of 3000 data points where the data-point at iteration $i$ were sampled uniformly at random (without replacement) from the MNIST dataset. Table 7 illustrates the training loss of each different method across the different stages. As Table 7 reveals that in the early stages all methods achieve training loss close to 0 by overfitting the model while in the latter stages `CSVRG` and `SVRG` are able to achieve much smaller loss in comparison to `SGD`. We remark that `CSVRG` is able to meet the latter goal with way fewer FO calls than `SVRG` as demonstrated in Table 7.

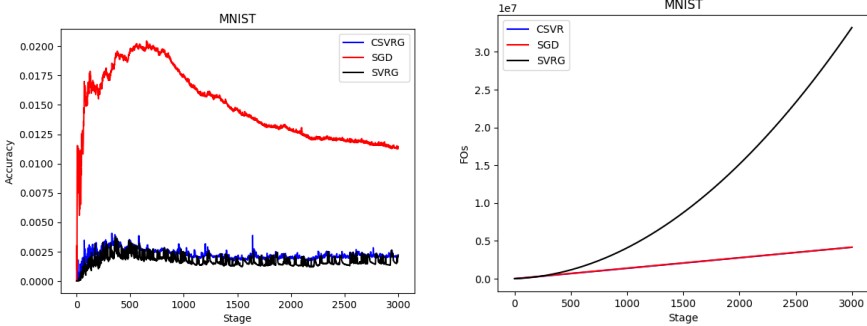

Table 7: Training loss as the stages progress on a neural network training for the MNIST dataset for the setting of sampling without replacement.

### K.2  DATA STREAMS WITH ASCENDING LABELS

Motivated by continual learning at which new tasks appear in a streaming fashion, we experiment with a sequence of MNIST data points at which *new digits* gradually appear. More precisely, we construct the stream of data points as follows:

1. For each of the classes $i \in \{0, 9\}$, we randomly sample 300 data points.

2. For the first 600 stages we randomly mix $0/1$ data points.

3. The stages $\{601 + (i-2)*300, 600 + (i-1)*300\}$ for $i \in \{2,9\}$ contain the data points of category $i$ (e.g. category 2 appears in stages $\{601, 900\}$).

Table 8 illustrates the training loss of CSVRG and SGD for the various stages of the above data-stream. As Table 8 reveals, CSVRG is able to achieve significantly smaller loss than SGD with the same number of FOs. We also remark that that the bumps appearing in bot curves corresponds to the stages at which a *new digit* is introduced.

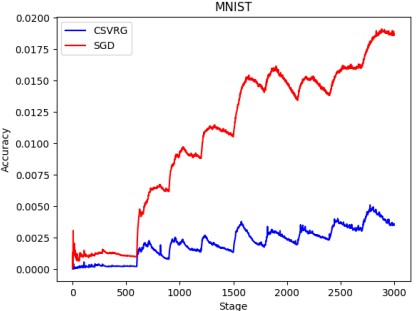

Table 8: The training loss of CSVRG and SGD for the various stages.

In Table 9 we present the classification accuracy of the models respectively produced by CSVRG and SGD for the various stages. At each stage $i$ we present the accuracy of the produced model according to the categories revealed so far. The sudden drops in the accuracy occur again at the stages where new digits are introduced. For example in iteration 601, the accuracy drops from roughly 1 to roughly 0.667 due to the fact in stage 600 the accuracy is measured with respect to the $0/1$ classification task while in stage 601 the accuracy is measured with respect to the $0/1/2$ classification task. At stage 601 both models misclassifies all the 2 examples and thus the accuracy drops to $2/3$. In the left plot, we plot the classification accuracy with respect to the data used in the training set while in the right one we plot the classification accuracy on unseen test data. Both plots reveal that CSVRG admits a noticable advantage of SGD.

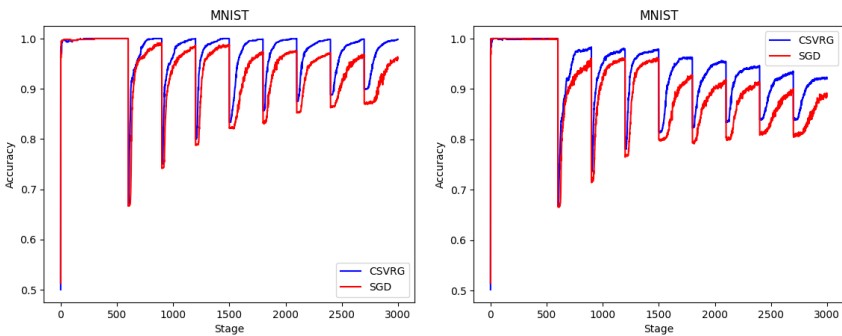

Table 9: Classification accuracy. For the train set on the left and the test set on the right. As a comparison between the two algorithms.

