# OpenReview forum: "Efficient Continual Finite-Sum Minimization"
_ICLR.cc/2024/Conference — ICLR 2024 poster_

### Official Review · Reviewer_FDce · 2023-10-28

**Soundness:** 3 good
**Presentation:** 4 excellent
**Contribution:** 3 good
**Rating:** 6
**Confidence:** 3

**Summary:**

This paper considers the instance-optimal finite-sum minimization problem, where the setting is we have a sequence of functions $f_1, \cdots, f_n$, and we need to output a sequence of points $x_1^*, \cdots, x_n^*$, with $x_i^*$ being the minimizer of $\sum_{j=1}^i f_j(x)/i$. This problem is inline with the modern machine learning setting as usually the training set is went through for only once, but we hope to make sure the current solution is the optimal for the current situation.

For this setting, the existing SGD or variance reduced methods are not optimal, because naively we need to call these methods $n$ times in order to find these $n$ minimizers. This paper proposed a new algorithm which finds the minimizer iteratively, which incurs $\tilde{O}(n/\epsilon^{1/3} + \log n /\sqrt{\epsilon})$ FO complexity. The authors also show that the lower bound of this problem is $\tilde{O}(n/\epsilon^{1/4})$, which means the newly proposed algorithm is close to optimal.

**Strengths:**

Originality: the setting is interesting, and the proposed algorithm is novel as far as I know. In their algorithm, they propose to switch between compute full gradient with $i$ FOs and compute a simple modification based on the previous gradient with 1 FO for full gradient estimation, which is a special trick.

Quality: The results are nice, because the authors not only provide the convergence guarantees, but also compare it with the existing methods, and provide a very close lower bound.

Clarity: The overall writing of this paper is excellent.

Significance: I think the results are interesting, because this setting is an important setting to explore, and they have presented a full story about this setting.

**Weaknesses:**

I guess the main weakness of this paper is about the strong convexity assumption of f. This is kind of old-school assumption, because nowadays we always use deep neural networks to fit the data, and these models are almost never strongly convex.  In fact, as far as I know, variance reduced techniques are not so useful for neural networks, so I am afraid that the proposed method are not very practical for real world models.

**Questions:**

I do not have additional questions. I have went through the proofs in the main paper, and they look correct to me. However, I did not get time to verify the long proofs in the appendix.

---

> ### Author Response · Authors · 2023-11-14
>
> Thank you very much for all the work and the positive feedback on our work.
>
> We just want to mention that the primary scope of our work was to formally frame and investigate the problem of efficient updates of models upon the arrival of new data. We agree with the reviewer that going beyond the strong convexity assumption is a very important and interesting research direction. Investigating the strong convex case serves as an important first step towards this direction while it already tackles several challenges and reveals surprising properties (especially with respect to lower bounds).
>
> We would also like to remark that we have added additional experimental evaluations for NNs (see Appendix K and our response to Reviewer jqUD and Reviewer s2LS) revealing interesting results for SIOPT-Grad.
>
> We hope that our work will instantiate a research direction towards investigating efficient update methods with favorable empirical and theoretical performance.

---

> > ### Comment · Reviewer_FDce · 2023-11-22
> >
> > I have read the rebuttal and will keep my score unchanged.

---

### Official Review · Reviewer_zvPG · 2023-10-31

**Soundness:** 3 good
**Presentation:** 3 good
**Contribution:** 3 good
**Rating:** 8
**Confidence:** 3

**Summary:**

This paper studies a new problem of efficient instance-optimal finite-sum minimization, where given a sequence of functions, the goal is to find a sequence of points $x_1^\star, \ldots, x_n^\star $ such that for each $i \in [n]$, point $x_i^\star$ approximately minimizes $\sum_{j=1}^{i} f_j(x) / i $, with the number of first-order oracle(FO) calls as least as possible. Given the condition that each function $f_j(\cdot)$ is strongly convex, the proposed method SIOPT-Grad produces an $\epsilon$-optimal sequence within $\tilde{O}(n/ \epsilon^{1/3} + 1/\sqrt{\epsilon})$ FOs. In addition, they prove a lower bound that, there is no \emph{natural} FO methods that completes this task within $O(n/\epsilon^{1/4}) queries.

In the regime motivated by the empirical risk minimization in the strongly convex case where statistical error is $\epsilon = O(1/n)$,  SIOPT-Grad requires only $O(n^{4/3})$ FOs compared to all previous FO solutions.

**Strengths:**

- The studied problem is well-motivated, both from the literature review on incremental learning, and from empirical estimation.
- The theoretical analysis is complete, an upper bound and lower bound is provided for this problem, as well as compared to state of the art as in table 1.
- The logic of this paper is easy to follow, and the assumptions/notations are presented in a clear way.
- The paper has additional experiments on the ridge regression problem.

**Weaknesses:**

- The upper bound provided by the algorithm is not tight compared to the lower bound.
- The algorithm only work in the strongly convex case.

Minor Issue:
- there is no input in the algorithm 2.
- the value of $\alpha$ needs to be in line 2 of algorithm 1.
- additional "the" in the second line of the first paragraph of section 3.1
- the VR is never defined. suggestion: "variance reduction(VR)" and then use VR afterwards

**Questions:**

- The solution proposed in this paper, although is better in the number of FOs, may result in much added overall running time. Is there any practical regime where the number of FOs is expensive, compared to the efficiency of the algorithm measured by the running time?

---

> ### Author Response · Authors · 2023-11-14
>
> Thank you for your work and the positive feedback on our paper. We will incorporate all the minor issues and thank you for spotting them.
>
> Concerning your question: We believe that considering *projection-free methods* that even with higher FO complexity may result in better running time due to the lack of a projection step, is a very interesting research direction.

---

> > ### Comment · Reviewer_zvPG · 2023-11-15
> >
> > Thanks for the reply, I maintain my score for this paper.

---

### Official Review · Reviewer_s2LS · 2023-11-01

**Soundness:** 3 good
**Presentation:** 3 good
**Contribution:** 3 good
**Rating:** 6
**Confidence:** 2

**Summary:**

The paper introduces the following natural problem: given a sequence of n convex functions. Find a sequence of inputs that minimizes the partial sum of the functions. Standard methods such as SGD require $O(n/\epsilon)$ first order gradient information about the functions to get error up to additive $\epsilon$. That paper introduces a novel algorithm which obtains the same accuracy using substantially fewer number of first order gradient information. In particular, the new algorithm only requires $O(n/\epsilon^{1/3})$  gradient calls. For small $\epsilon$, this is a big improvement. For instance, the authors note that $\epsilon=1/n$ and is a very common setting, in which case their algorithm gives a polynomial improvement. The authors also introduce the notion of a 'natural algorithm' and prove a lower bound of $\Omega(n/\epsilon^{1/4})$ gradient calls under the assumption.

**Strengths:**

The introduction of the problem is a nice conceptual contribution. The new algorithm they proposed could also have other applications. In the notion of a "natural algorithm" is a nice contribution since it allows the analysis of algorithms and lower bounds.

**Weaknesses:**

A weakness is the lack of intuition about their algorithm. I mostly follow the math, however conceptually I do not know why exactly they can get an improvement in the epsilon power. It seems like the high level idea is only to compute a gradient update if we have not had an update for a long time.

Establishing that the gradient is unbiased seems fairly straightforward: it just uses the fact that the gradient at the next step is a linear combination of the new function and the previous gradient. Analyzing the variance seems like some technical work. However I did not understand the conceptual improvement.

Can the authors comment more about the intuition of their algorithm?

Another weakness is the experimental setting. In the experiments, the authors analyze a version of ridge regression. They divide by $i$ for every step which seems a bit unnatural to me. It would make more sense if the $i$ term was not there. But in that case, the problem becomes very easy. By using standard rank one update formulas, the *exact* optimum can be found in $O(nd^2)$ time. Perhaps the authors divided by $i$ to make the problem `artificially' harder? In any case, testing the algorithm on other natural problems which cannot be solved by other methods would be more convincing. The authors also do not list the parameters of the dataset and the $\epsilon$ used in the algorithm.

**Questions:**

Please see above.

---

> ### Author Response · Authors · 2023-11-14
>
> We would like to thank the reviewer for their work and comments.
>
> Up next we provide the intuition behind our algorithm in $3$ main steps.
>
> - **Main Intuition of VR methods**
>
>   The main idea of VR methods is to compute a full-gradient at an anchor point $\tilde{x}$, $\tilde \nabla := \frac{1}{n}\sum_{j=1}^n \nabla f_j(\tilde x)$ and use the following unbiased gradient estimator.
>   $$\nabla_t = \nabla f_j(x_t)- \nabla f_j(\tilde x) + \tilde \nabla \text{ with }j \sim \mathrm{Unif}(1,\ldots,n)$$
>
>   The main idea behind this estimator is that *the closer $\tilde{x}$ is to the optimal point $x^\star$ the smaller the variance of $\mathbb{E}\left[||\nabla_t - \nabla g(x_t)||^2\right]$* where $g(x) = \sum_{j=1}^nf_j(x)/n$.
>
>   In order to understand the latter let $\tilde{x} := x^\star$. Initializing the above process for $x^0 = x^\star$ results in $\nabla_0= \nabla g(x^\star)$ meaning that the method stands still on the optimal point. On the contrary SGD even initialized at $x^\star$ will move before converging back in general (e.g., without the strong growth setting).
>
> - **Straight-forward Approach in Instance-Optimal Finite-Sum Minimization**
>
>   In order to tackle Problem $2$, probably the most intuitive idea is the following. During stage $i \in [n]$, set the anchor point $\tilde x_i = \hat x_{i-1}$ (the previous output of the method) and compute $\tilde \nabla_i = \frac{1}{i}\sum_{j=1}^i \nabla f_j(\tilde x_i)$. Then during stage $i \in [n]$ perform SGD starting from the point $x^0_i := \hat x_{i-1}$ and use the estimator
>   $$\nabla_t^i = \nabla f_j(x_t)- \nabla f_j(\tilde{x}) + \tilde{\nabla}_i \text{ with }j \sim \mathrm{Unif}(1,\ldots,n).$$
>
>   The intuition behind this approach is that for large values of $i$, $\hat{x}_{i-1}$ is close to $x^\star_i$ (optimal solution of stage $i$) which implies that the variance reduction mechanism that we described above is activated leading to accelerated rates.
>
>   To understand why $\hat x_{i-1}$ is close to $x^\star_i$, notice that $||x_{i-1}^\star - x_{i}^\star|| \leq O(1/i)$ (see Lemma $6$) while we can inductively assume that $||\hat x_{i-1} - x_{i-1}^\star|| \leq \epsilon$.
>
>   The caveat of this approach is that computing $\tilde \nabla_i = \frac{1}{i}\sum_{j=1}^i \nabla f_j(\hat x_{i-1}) $ at each stage $i \in [n]$ results in overall $O\left(\sum_{i=1}^n i\right) = O(n^2)$ FOs.

---

> > ### Author Response · Authors · 2023-11-14
> > **Continuation of the Comment**
> >
> > - **Our approach**
> >
> >   In order to reduce the $O(n^2)$ FOs of the previous approach, at stage $i \in [n]$, instead of using $\hat x_{i-1}$ as the anchor point of stage $i$, we use $\hat x_{\text{prev}}$ where $\text{prev} \ll i-1$ for most stages $i$. In other words we compute $\tilde \nabla_i = \frac{1}{i}\sum_{j=1}^i f_j(\hat x_{\text{prev}})$. The advantage of this approach is that updating $\tilde \nabla_i$ can be performed with only $1$ additional FO at each stage,
> >   $$\tilde \nabla_i = \left(1 - \frac{1}{i}\right)\cdot \tilde \nabla_{i-1} + \frac{1}{i} \cdot \underbrace{\nabla f_i (x_{\text{prev}})}_{\text{new FO}}~~~~~\text{(Step 16 of Algorithm 1)}.$$
> >
> >   During stage $i \in [n]$, we perform $T_i$ steps of SGD starting form $x_i^0 = \hat x_{i-1}$ and using the unbiased estimator
> >   $$\nabla_i^t := (1-\frac{1}{i}) (\nabla f_{u_t}(x_t) - f_{u_t}(x_{\text{prev}}) + \tilde \nabla_{i-1})  + \frac{1}{i}\cdot \nabla f_i(x_i^t)~~\text{(Step 5 in Algorithm 2)}$$
> >   where $\tilde \nabla_{i-1} = \frac{1}{i-1}\sum_{j=1}^{i-1} \nabla f_j(\hat x_{\text{prev}})$.
> >
> >   As in the case of the standard esimator of VR methods, the above estimator also admits the property that *the closer $\hat x_{\text{prev}}$ and $\hat x_{i-1}$ are to the optimal solution $x^\star_i$, the smaller the variance of the estimator*. To understand the latter notice that in case $\hat x_{\text{prev}} = \hat x_{i-1} = x_i^\star$ then Algorithm $2$ stands still to $x_i^\star$.
> >
> >   The final step of intuition behind our algorithm consists of understanding why $\hat x_{\text{prev}}$ is close enough to $x_{i}^\star$ (the fact that $\hat x_{i-1}$ is close to $ x_i^\star$ has already been explained). In order to minimize the overall number of FOs that SIOPT-Grad uses, $\text{prev}$ is sparsely updated. Namely, $\text{prev}$ is updated in stages $i$ for which $i - \text{prev} \geq \alpha\cdot i$ (Step $4$ of Alg $1$).
> >
> >   On first sight, the latter implies that the larger $i$ is, the larger the distance between $\hat x_{prev}$ and $x^\star_i$ will be - something that **does not** serve our purposes. Fortunately, the latter concern is not the case. More precisely, Lemma $6$ reveals that,
> >   $$||\hat x_j-x_i^\star||^2_2\leq \frac{8}{\mu^2}\left(\frac{G(i-j)}{i+j}\right)^2+2||\hat x_j - x^\star_j||^2_2~~\text{for } j \in [i-1].$$
> >   At an intuitive level, Lemma 6 establishes that as $i$ increases the more *similar* the optimal points $x_i^\star$ become. More precisely, since $i - prev < \alpha \cdot i$, we are ensured that
> >   $$||\hat x_{\text{prev}}-x_i^\star||^2_2\leq \frac{8}{\mu^2}\left(G \alpha \right)^2+2||\hat x_{prev} - x^\star_{prev}||^2_2 \leq \frac{8}{\mu^2}\left(G\alpha \right)^2+\frac{2\epsilon^2}{\mu^2}$$
> >   where the last inequality comes by inductively assuming that $g_{prev}(\hat x_{prev}) - g_{prev}(x^\star_{prev}) \leq \epsilon$. To this point it is clear that by selecting $\alpha$ sufficiently small, results in $\hat x_{prev}$ and $\hat x_i$ being close enough resulting in the activation of the variance reduction mechanism.
> >
> >   We hope the above discussion clarifies the intuition behind our algorithm. We plan to incorporate the above discussion in the revised version of our paper.
> >
> >
> > Regarding our experimental section.
> >
> >
> > - We would like to clarify that at each stage $i \in [n]$ dividing with $i$ captures statistical error of the prefix of the first $i$ samples. For the exact same reason, VR methods divide the finite-sum function with $n$ (e.g. see [1]).
> >
> > - We used ridge regression for two main reasons, the first one being that it satisfies all of the assumptions made in our theoretical proofs and therefore we could demonstrate our theoretical results in practice. The second reason is exactly because in this setting there are linear algebra solvers that allow us to calculate the optimal solutions and therefore we can display the suboptimality gap of the various methods. The list of parameters used in experimental evaluations are presented in Appendix I.
> >
> > - In Appendix K we have added additional experimental evaluations for 2-layer Neural Networks in the continual learning example on the MNIST data set that we described above, which illustrate the advantages of SIOPT-Grad (see also our response to Reviewer jqUD).
> >
> > [1] Katyusha: The First Direct Acceleration of Stochastic Gradient Methods, Allen-Zhu 2018
> >
> >
> > We hope that the above alleviate your concerns and will make you reconsider your score. We remain at your disposition for further clarifications and discussion.

---

> > > ### Author Response · Authors · 2023-11-15
> > > **Discussion about Neural Network Experiments**
> > >
> > > In order to facilitate further discussion we attach some new plots from Appendix K of the revised version of the manuscript that display the performance of SIOPT on training 2-layer Neural Networks.
> > >
> > > We present two categories of plots. The first one addresses the same setting as our previous experiments for ridge regression, where you draw samples from a stationary distribution and at each stage a new sample is presented to the model.
> > >
> > >  [Stationary Distributions](https://imgur.com/a/h9TZmFu)
> > >
> > > On the rest of the plots we measure the performance of our method on data streams with ascending labels, a problem setting motivated by continual learning. Here in the early stages we start from a binary classification problem (0s and 1s on MNIST) and then as time progress new classes are introduced to the dataset (digits 3, 4, 5, etc). SIOPT outperforms SGD in this setting and achieves smaller training loss at each stage. It is also interesting to point out that after the introduction of a new class of labels SIOPT manages to fit the model faster into the new problem with respect to the number of stages that need to pass, in comparison to SGD and maintain higher accuracy across all stages.
> > >
> > > [Ascending Labels](https://imgur.com/a/BWy8TVR)
> > >
> > > [Ascending Labels Train Accuracy](https://imgur.com/a/HEJKxlv)
> > >
> > > [Ascending Labels Test Accuracy](https://imgur.com/a/YbSVEno)

---

> ### Comment · Reviewer_s2LS · 2023-11-19
> **Response to rebuttal**
>
> Thank you to the others for their response. I would encourage them to add some of the intuition in any future version of the paper.
>
> The new MNIST experiments are quite interesting. It is nice how the performance of SI-OPT is always an upper envelope to the performance of SGD in the ascending labels experiment.
>
> I read the review of reviewer Reviewer jqUd. It seems like they are quite knowledgeable about grading descent based algorithms. So I will momentarily hold off on any updates until they have a chance to respond about their proposed algorithm. I see how the exact stated algorithm does not work. But it is stated in quite informal terms, so maybe it can be easily modified. Let's give them a chance to respond.

---

> > ### Author Response · Authors · 2023-11-20
> >
> > We would like to thank the reviewer for their comments. We have uploaded a new revised version of the work that includes:
> >
> > 1. The MNIST experiments as well as a discussion on them.
> >
> > 2. The additional intuition as we explained it in our previous reply.
> >
> > In the revised version we have also incorporated the minor comments from the rest of the reviewers.
> >
> > To this end we would like to verify whether your concerns about the intuition of the algorithm and the additional experiments are covered in this updated manuscript.
> >
> > We remain at your disposal for any further questions.

---

> > ### Author Response · Authors · 2023-11-22
> >
> > We want to mention that reviewer jqUd has replied to our rebuttal without any objections to our comments.
> >
> > Therefore we would like to thank you for all your work and we hope that the above in conjunction with our comments and the revised manuscript will help you reconsider your score.

---

> > > ### Comment · Reviewer_s2LS · 2023-11-23
> > > **Follow up**
> > >
> > > Thanks, I will update my score.

---

### Official Review · Reviewer_jqUd · 2023-11-06

**Soundness:** 3 good
**Presentation:** 3 good
**Contribution:** 2 fair
**Rating:** 5
**Confidence:** 4

**Summary:**

The paper studies the following problem: given $n$ functions, the goal is to find $n$ points such that the $i$-th point minimizes the average of the first $i$ functions. For this problem, the paper presents a lower bound on the number of iterations for a certain class of algorithms, called natural first-order methods. They present an algorithm that achieves first-order oracle complexity which is close to the lower bound.

**Strengths:**

A clearly written paper with sound results.

**Weaknesses:**

I have the following concerns about the paper:

-- I can’t connect Problem 2 with its motivation. For example, you say that “it is important that a model is constantly updated so as to perform equally well both on the past and the new data”, but:
1) Problem 1 achieves precisely that;
2) in Problem 2, you train *multiple* models, with later models performing well on all data, and with older models not taking into account new data at all.

To conclude, I don’t see a motivation for the problem.

-- You give lower bounds only for natural algorithms. If you don’t restrict yourself to natural algorithms (and it’s not clear to me why you consider only natural algorithms), I believe that $O(\frac{\log n}{\epsilon})$ bound is possible, which outperforms all suggested methods. The idea is to run the following simple algorithm:

find $\epsilon$-minimizer $x_n$ of $g_n$ using SGD
for i = n-1, n-2, …, 1:
	starting from $x_{i+1}$, find $\epsilon$-minimizer $x_i$ of $g_i$ using SGD

I didn’t check the math in detail, but I believe that the proof goes like this:
1) If $x_{i+1}$ is $\epsilon$-minimizer of $g_{i+1}$, then $x_{i+1}$ is $(1 + 1/i) \epsilon$-minimizer of $g_i$.
2) The number of iterations required to find $x_i$ starting from $x_{i+1}$ is $O(\frac{\log (1 + 1/i)}{\epsilon})$
3) Summing this over all $i$, we get $O(\frac{\log n}{\epsilon})$ iterations

-- Experiments are only conducted in simple regression settings. Dataset statistics (e.g. the number of points) are not shown. It is unclear how exactly you execute the baselines.

**Questions:**

Minor issues:

-- Commas are missing in many places

-- You mention $O(n \log \frac{1}{\epsilon})$ complexity for VR methods, but isn’t the same bound achievable by trivial deterministic gradient descent, by sampling all $n$ functions at every iteration?

-- Page 2: “for Problem 1 using $O(1/\epsilon)$.” - “FOs” is missing.

---

> ### Author Response · Authors · 2023-11-14
>
> We thank the reviewer for the work and the valuable comments.
>
> - *"I can’t connect Problem 2 with its motivation. For example ... past and the new data”*
>
>   We believe that some confusion might have been created due to the use of $n$ in both for Problem $1$ and Problem $2$. In many cases  the *overall $n$ data points are not available* from the beginning and are rather gathered in an online fashion as the system is deployed to the public.
>
>   We agree with the reviewer that if we had access to all $n$ data points from the beginning, we could directly solve Problem $1$. However our work is motivated from settings where only a small fraction of $n_1$ data points is available in the beginning and most $n-n_1$ data points are gradually revealed (for simplicity we set $n_1 =1$). To avoid confusion, we will use the notation $n_1$ to denote the initial data corresponding to Problem $1$.
>
>   To further illustrate the motivation of our work in the next item we provide a concrete example in the context of continual learning.
>
>
> - *Concerning the motivation of our work*
>
>   Efficiently updating a model upon the arrival of *"fresh data''* is a fundamental problem in ML. For example in *continual learning* models are initially trained with an initial set of data and then they are constantly updated so as to incorporate new tasks/data that arrive online. To clarify things let us provide a continual learning point of view with respect to the MNIST data set.
>
>   Let us assume that we are initially given $n_1$ data points from MNIST with only $0$ and $1$ labels. The initial goal is to train a model with parameters $x$ to correctly classify $0/1$ digits. As a result, one needs to solve the following finite-sum problem (Problem $1$)
>
>   $$\min_x \sum_{j=1}^{n_1} f_j(x)/{n_1}~~\text{where }f_j(\cdot) \text { captures the respective loss of data point j.} $$
>
>   After training the model (solving Problem $1$), an additional data point arrives with label $2$ (additional task). Our goal now is to update $x$ so that our model correctly classifies the $2$-example while preserving its classification ability for $0/1$'s. Thus we need to solve the following similar but different finite-sum problem
>   $$\min_x \sum_{j=1}^{n_1+1} f_j(x)/(n_1+1).$$
>
>   Now imagine the above process to be repeated for a stream of $n_2$ data points of label $2$. Namely, after each model-update a new $2$-data point arrives that then needs to be incorporated in the model. As a result, for each $i=1,\ldots,n_2$, we need to compute $ \hat x_i$ such that
>   $$ \sum_{j=1}^{n_1 + i} f_j(\hat x_i)/(n_1+i) -   \min_x \sum_{j=1}^{n_1+i} f_j(x)/(n_1+i) \leq \epsilon.$$
>   The latter can continue for streams of $n_3,n_4,\ldots,n_9$ data points (corresponding to the respective labels). However resolving Problem $1$ from stratch (even with a state-of-the art VR method) every time a new data point arrives leads to a huge computational burden. That is the reason Problem $2$ asks for *sequence of approximate solutions* with mimimum number of overall FO calls.
>
>   We remark that for the clarity of presentation, in Problem $2$ we set $n_1 = 1$ however our framework is directly extendable for general $n_1$. Moreover for the clarity of presentation we also focus on the case of a new data point arriving at each stage $i$, however our setting can be directly extended to the case where several data points arrive at each stage.
>
> - *Concerning the analysis of the algorithm that you propose*
>
>    1. *If $x_{i+1}$ is $\epsilon$-minimizer of $g_{i+1}$ then it is an $(1 + 1/i)\epsilon$-minimizer of $g_i$.*
>
>        We must be missing some hidden assumptions in your statement as the above statement is simply not true. Let $\epsilon =0$ then the above statement says that a minimizer of $g_{i+1}(x)$ is also a minimizer of $g_i(x)$. That is clearly not the case: e.g., if you solve the very first data term to full accuracy, then you do not need to care about the incoming data in the future.
>
>    2. *The number of iterations required to find $x_i$ starting from $x_{i+1}$ is $O\left(\frac{ \log (1 + 1/i)}{\epsilon}\right)$.*
>
>        We do not see how the above expression is derived. Please clarify if you make some assumptions in order to reach these conclusions.
>
>        We remark that even GD initialized at $x_0$ requires $O(\log ( ||x_0 - x^\star||/\epsilon))$ iterations, with each iteration requiring $i$ FOs.
>
>        We additionally remark that SGD does not admit such *warm-start* guarantees (even outside log), $O( (f(x_0) - f(x^\star))/\epsilon)$ in general. The reason is that such guarantees would imply that SGD initialized at $x_0 := x^\star$ would stand still with probability $1$ that is clearly not the case.

---

> > ### Author Response · Authors · 2023-11-14
> > **Continuation of the Comment**
> >
> > - *Concerning our experimental evaluations*
> >
> >   We first want to remark that our work is mainly theoretical. Thus in our experiments we chose settings matching the requirements of our theoretical results. We chose ridge regression at which the exact optimal solution can be analytically computed so as to be able to precisely track the suboptimality of each baseline.
> >
> >   In our experimental evaluation, we reveal a new data point at each stage $i$ that appear in the $x$-axis of our plots. All the baselines are implemented according to the presented algorithms in the respective papers and the parameters used are presented in Appendix I.
> >
> >   We remark that in Appendix K we have added additional experimental evaluations for $2$-layer Neural Networks in the continual learning example on the MNIST data set that we described above, which illustrate the advantages of SIOPT-Grad.
> >
> > - *Questions*
> >
> >      - Thank you for the suggestion. We will fix the missing commas.
> >
> >      - GD requires $O( n \frac{L}{\mu} log (1/\epsilon) )$ but VR methods decouple the $n$ and $L/\mu$. For example Katyusha admits $O( n \log (1/\epsilon) + \sqrt{n\frac{L}{\mu}} \log(1/\epsilon) )$ that is quite beneficiary. We will clarify in the revised version.
> >
> >      - Thank you for spotting the typo.
> >
> >
> > We hope that the above alleviate your concerns and will make you reconsider your score. We remain at your disposition for further clarifications and discussion.

---

> ### Author Response · Authors · 2023-11-15
> **Discussion about Neural Network Experiments**
>
> In order to facilitate further discussion we attach some new plots from Appendix K of the revised version of the manuscript that display the performance of SIOPT on training 2-layer Neural Networks.
>
> On the first plot we display the performance of our method on data streams produced by stationary distributions. As we see in the ridge regression case our method achieves performance similar to SVRG with much fewer FOs.
>
>  [Stationary Distributions](https://imgur.com/a/h9TZmFu)
>
> On the rest of the plots we present the performance of our method on data streams with ascending labels, a problem setting motivated by continual learning. In this problem we initially wish to classify two labels (0s and 1s on MNIST) and then as time progress new classes are introduced to the dataset (digits 3, 4, 5, etc), this setting is similar to the one we discussed above as motivation for our work. SIOPT here achieves higher performance than SGD and adapts to the new tasks faster.
>
> [Ascending Labels](https://imgur.com/a/BWy8TVR)
>
> [Ascending Labels Train Accuracy](https://imgur.com/a/HEJKxlv)
>
> [Ascending Labels Test Accuracy](https://imgur.com/a/YbSVEno)

---

> ### Author Response · Authors · 2023-11-21
> **End of Discussion**
>
> As the discussion period is concluding tomorrow, we would like to ask the reviewer if our replies have covered all of your concerns about our work.
>
> We remain at your disposal for any questions.

---

> > ### Comment · Reviewer_jqUd · 2023-11-22
> >
> > Thank you for your response, I appreciate the feedback and would like to keep the original score.

---

### Author Response · Authors · 2023-11-17
**Summary of Authors replies**

Dear Reviewers,

We thank you all for your time and helpful reviews. Up next we summarize some key points from our replies:

- We have updated the draft with additional experiments on 2-layer neural network. The experiments can be found in appendix K of the draft.

- In our reply to reviewer jqUd we provided an extended motivation and explanation of our setting.

- In our reply for reviewer s2LS we provided the intuition behind the mathematical construction of our algorithm.

We remain at your disposal for further discussion and questions.

---

### Meta-Review · Area_Chair_XSuA · 2023-12-04

**Metareview:**

The paper considers the problem of finding a point that minimizes the average of j functions.  The problem is natural and well-motivated. The reviewers generally felt the paper had nice conceptual contributions.  The main weakness is assuming strong convexity for the results to hold.  Still, this paper has a significant contribution of interest to the community.

**Justification For Why Not Higher Score:**

The results are solid, but not particularly novel as such, I think this is a poster.

**Justification For Why Not Lower Score:**

Several reviewers felt the paper had a simplistic elegance to it for an important problem. I think the paper should be published to the community.

---

### Decision · Program_Chairs · 2024-01-16

Accept (poster)